# Causal Imitation for Markov Decision Processes: a Partial Identification Approach

**Kangrui Ruan**[*]
Columbia University
kr2910@columbia.edu

**Junzhe Zhang**[*]
Syracuse University
jzhan403@syr.edu

**Xuan Di**
Columbia University
sharon.di@columbia.edu

**Elias Bareinboim**
Columbia University
eb@cs.columbia.edu

## Abstract

Imitation learning enables an agent to learn from expert demonstrations when the performance measure is unknown and the reward signal is not specified. Standard imitation methods do not generally apply when the learner and the expert's sensory capabilities mismatch and demonstrations are contaminated with unobserved confounding bias. To address these challenges, recent advancements in causal imitation learning have been pursued. However, these methods often require access to underlying causal structures that might not always be available, posing practical challenges. In this paper, we investigate robust imitation learning within the framework of canonical Markov Decision Processes (MDPs) using partial identification, allowing the agent to achieve expert performance even when the system dynamics are not uniquely determined from the confounded expert demonstrations. Specifically, first, we theoretically demonstrate that when unobserved confounders (UCs) exist in an MDP, the learner is generally unable to imitate expert performance. We then explore imitation learning in partially identifiable settings — either transition distribution or reward function is non-identifiable from the available data and knowledge. Augmenting the celebrated GAIL method (Ho & Ermon, 2016), our analysis leads to two novel causal imitation algorithms that can obtain effective policies guaranteed to achieve expert performance.

## 1 Introduction

Children often learn how to behave in an unfamiliar environment by imitating adults. Imitation learning (IL) enables a learning agent to behave in an unknown environment by observing expert demonstrations. It provides a viable approach for policy learning from demonstrations when the reward function is not fully known and reward signals are not specified [2, 30, 8, 20, 31]. Imitation learning has been widely applied across disciplines, such as autonomous driving [13, 39], robotics [18], natural language processing [10, 11, 40, 41], and chronic disease management [52, 44].

It has been acknowledged in the literature that imitation learning could face significant challenges when *unobserved confounding bias* in expert demonstrations cannot be ruled out *a priori* [17, 57, 27, 42]. For illustration with simplicity, consider a Multi-Armed Bandit (MAB) model [28] described in Fig. 1; $X \in \{0, 1\}$ is a binary action, and

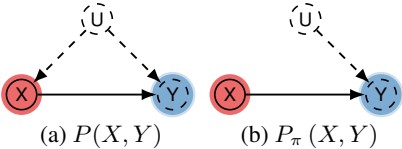

Figure 1: A multi-armed bandit model.

[*]Equal contribution; Kangrui: work performed while at Columbia University.

|  | **Identifiable Reward** | **Non-Identfiable Reward** |
|---|---|---|
| **Identifiable Transition** | Standard IRL (e.g., GAIL[19]) | CAIL-$\mathcal{R}$ (Alg. 1 in Sec. 3.1) |
| **Non-Identfiable Transition** | CAIL-$\mathcal{T}$ (Alg. 2 in Sec. 3.2) | Inimitable (Thm. 1 in Sec. 2.1) |

Table 1: Summary of main contributions in this paper, including the analysis and proposed algorithms.

$Y$ is the reward; $U$ is a latent covariate (to the imitator) uniformly drawn over a binary domain $\{0, 1\}$. Values of the reward are decided by a reward function $Y \leftarrow X \oplus U$ where $\oplus$ is a "*xor*" operator. An expert demonstrator, having access to covariate $U$, selecting action based on an expert policy $X \leftarrow \neg U$. Evaluating the expert's performance gives $\mathbb{E}[Y] = \mathbb{E}[\neg U \oplus U] = 1$. On the other hand, an imitator, mimicking the expert's behavior, will follow a policy $\pi(X) = P(X) = 0.5$, selecting action uniformly at random. Evaluating the imitator's performance gives $\mathbb{E}_\pi[Y] = \sum_x \pi(x)\mathbb{E}[x \oplus U] = 0.5$, far from the expert's performance $\mathbb{E}[Y] = 1$.

Causal Inference (CI) addresses the challenges of unobserved confounding bias within the observational data [32, 47, 53]. It leverages causal knowledge integral to the data generation process, typically represented as a causal diagram or potential outcomes [32, 43, 5]. More recently, incorporating causal inference methods into the imitation learning paradigm, *causal imitation learning* has evolved into a critical area of research [16, 57, 27, 7, 49, 42]. To compensate for the presence of unobserved confounding bias, these methods rely on additional structural or parametric knowledge about causal relationships among variables in the environment. By utilizing such domain knowledge, the imitator is able to recover the underlying system dynamics (i.e., causal effect) from confounded expert demonstrations and, in turn, obtain an imitating policy that can achieve the expert's performance.

By and large, the combination of causal knowledge and observational data does not always allow one to point-identify the causal effect, called the *non-identifiable*. That is, more than one parametrization of the target effect is compatible with the same observational data and model assumptions [32, Def. 3.2.2]. For instance, in the MAB environment described previously, the imitator's performance $\mathbb{E}_\pi[Y]$ is not identifiable from the confounded observational distribution $P(X, Y)$ [32, Thm. 3.4.1]. *Partial identification* methods concerned with inferring about target causal effects in non-identifiable settings, and has been a target of growing interest in the domains of causal inference [3, 12, 37, 14, 58], econometrics [21, 35, 38, 48], and more recently, in machine learning [25, 24, 23]. Among these works, two approaches are often employed: (1) bounds are derived for the target effect under minimal assumptions, or (2) additional untestable assumptions are invoked under which the causal effect is identifiable, and then sensitivity analysis is conducted to assess how the target causal effect varies as the untestable assumptions are changed. Despite their effectiveness in addressing data bias, partial identification has still been rarely explored in the context of imitation learning.

This paper studies the partial identification for imitation learning in a generalized sequential decision-making environment of Markov Decision Processes (MDPs, [36]). The imitator must determine a sequence of actions, while unobserved confounders cannot be ruled out a priori in expert demonstrations. We discuss the solutions case-by-case, depending on the identifiability of the underlying system dynamics from the confounded data, including the reward function $\mathcal{R}$ and the transition distribution $\mathcal{T}$. Specifically, our contributions can be summarized as follows. (1) We theoretically prove that when unobserved confounders generally exist, it is infeasible to learn a robust policy that is guaranteed to achieve expert performance from the demonstration data. (2) When only the transition distribution $\mathcal{T}$ is identifiable, we propose a novel imitation algorithm that leverages the bounds over the non-identifiable reward $\mathcal{R}$; by matching the weighted occupancy measure, the imitator is able to obtain a policy that can outperform the expert. (3) We propose an alternative algorithm when the reward $\mathcal{R}$ is identifiable, but there is unobserved confounding affecting the transition $\mathcal{T}$. Our proposed algorithms could be implemented by augmenting the celebrated generative adversarial imitation learning framework (GAIL, [19]). Table 1 briefly summarizes this paper's main contributions. Due to space constraints, all proofs are provided in Appendices A and B.

## 1.1 Preliminaries

This section introduces the basic notations and definitions used throughout the paper. We use capital letters to denote random variables ($X$), small letters for their values ($x$), and $\mathscr{D}_X$ for the domain

of $X$. For an arbitrary set $\boldsymbol{X}$, let $|\boldsymbol{X}|$ be its cardinality. Fix indices $i, j \in \mathbb{N}$. Let $\boldsymbol{X}_{i:j}$ stand for a sequence of variables $\{X_i, X_{i+1}, \ldots, X_j\}$; for consistency, the sequence $\boldsymbol{X}_{i:j} = \emptyset$ if $j < i$. We denote by $P(\boldsymbol{X})$ represents a probability distribution over variables $\boldsymbol{X}$, and will consistently use $P(\boldsymbol{x})$ as abbreviations for probabilities $P(\boldsymbol{X} = \boldsymbol{x})$. Finally, $\mathbb{1}\{\boldsymbol{Z} = \boldsymbol{z}\}$ is an indicator function that returns 1 if event $\boldsymbol{Z} = \boldsymbol{z}$ holds true; otherwise, it returns 0.

The basic semantical framework of our analysis rests on *structural causal models* (SCMs) [32, 4]. An SCM $M$ is a tuple $\langle \boldsymbol{V}, \boldsymbol{U}, \mathcal{F}, P(\boldsymbol{U}) \rangle$, where $\boldsymbol{V}$ is a set of endogenous variables and $\boldsymbol{U}$ is a set of exogenous variables. $\mathcal{F}$ is a set of functions s.t. each $f_V \in \mathcal{F}$ decides values of an endogenous variable $V \in \boldsymbol{V}$ taking as argument a combination of other variables in the system. That is, $V \leftarrow f_V(\boldsymbol{PA}_V, \boldsymbol{U}_V), \boldsymbol{PA}_V \subseteq \boldsymbol{V}, \boldsymbol{U}_V \subseteq \boldsymbol{U}$. Exogenous variables $U \in \boldsymbol{U}$ are mutually independent, values of which are drawn from the exogenous distribution $P(\boldsymbol{U})$. Naturally, $M$ induces a joint distribution $P(\boldsymbol{V})$ over endogenous variables $\boldsymbol{V}$, called the *observational distribution*. An intervention on a subset $\boldsymbol{X} \subseteq \boldsymbol{V}$, denoted by $\mathrm{do}(\boldsymbol{x})$, is an operation where values of $\boldsymbol{X}$ are set to constants $\boldsymbol{x}$, replacing the functions $\{f_X : \forall X \in \boldsymbol{X}\}$ that would normally determine their values. For an SCM $M$, let $M_{\boldsymbol{x}}$ be a submodel of $M$ induced by intervention $\mathrm{do}(\boldsymbol{x})$. For a set $\boldsymbol{Y} \subseteq \boldsymbol{V}$, the interventional distribution $P_{\boldsymbol{x}}(\boldsymbol{Y})$ induced by $\mathrm{do}(\boldsymbol{x})$ is defined as the joint distribution over $\boldsymbol{Y}$ in the submodel $M_{\boldsymbol{x}}$, i.e., $P_{\boldsymbol{x}}(\boldsymbol{Y}; M) \triangleq P(\boldsymbol{Y}; M_{\boldsymbol{x}})$. We leave $M$ implicit when it is obvious from the context. For a detailed survey on SCMs, we refer readers to [32, Ch. 7] and [4].

## 2 Challenges of Unobserved Confounding

We focus on the sequential decision-making setting of an agent operating in a MDP environment [36] over a series of interventions $t = 1, 2, \ldots$. At each time step $t$, the agent observes the current state $S_t$, performs an action $\mathrm{do}(X_t)$, receives a subsequent reward $Y_t$, and moves to the next observed state $S_{t+1}$. Values of the action $X_t$ are selected by sampling from a stationary policy $\pi(x \mid s)$, which is a function mapping from the domain of the observed state $S_t$ to the probability space over the domain of every action $X_t$. Let $\boldsymbol{U}_t$ be an unobserved noise independently drawn from an exogenous distribution $P(\boldsymbol{U})$. Values of the subsequent reward $Y_t$ and the next state $S_{t+1}$ are, respectively, determined by structural functions $y_t \leftarrow f_Y(s_t, x_t, \boldsymbol{u}_t)$ and $s_{t+1} \leftarrow f_S(s_t, x_t, \boldsymbol{u}_t)$, taking as input the current state $S_t$, action $X_t$, and latent noise $\boldsymbol{U}_t$. The initial state $S_1$ is drawn from an initial distribution $P(S_1)$. We will consistently use $\mathcal{X}, \mathcal{S}$, and $\mathcal{Y}$ to denote the domain of action $X_t$, state $S_t$, and reward $Y_t$. Like a standard discrete MDP, domains of actions $\mathcal{X}$ and states $\mathcal{S}$ are assumed to be finite; rewards are bounded in a real interval $\mathcal{Y} \triangleq [0, 1] \subset \mathbb{R}$. Naturally, the agent operating in this environment defines an interventional distribution $P_\pi$, summarizing the consequences of its actions.

Fig. 2a shows a graph describing this generative process; where nodes represent observed variables and directed arrows represent the functional relationships between them. For every time step $t > 1$, the current state $S_t$ "blocks" all pathways from previous nodes (e.g., $S_{t-1}$) to the future nodes (e.g., $S_{t+1}$) [32, Def. 1.2.3]. Applying d-separation rules leads to the following independence.

**Definition 1** (Markov Property [36]). For a joint distribution $P_*$ over sequences of states $S_1, S_2, \ldots$, actions $X_1, X_2, \ldots$, and rewards $Y_1, Y_2, \ldots$, the Markov property holds with regard to distribution $P_*$, if for every time step $t = 1, 2, \ldots, \left( \bar{\boldsymbol{S}}_{1:t-1}, \bar{\boldsymbol{X}}_{1:t-1}, \bar{\boldsymbol{Y}}_{1:t-1} \perp\!\!\!\perp \bar{\boldsymbol{X}}_{t:\infty}, \bar{\boldsymbol{S}}_{t+1:\infty}, \bar{\boldsymbol{Y}}_{t:\infty} \mid S_t \right)$.

It follows from Def. 1 that for any horizon $T$, the joint distribution $P_\pi \left( \bar{\boldsymbol{X}}_{1:T}, \bar{\boldsymbol{S}}_{1:T}, \bar{\boldsymbol{Y}}_{1:T} \right)$ generated by a policy $\pi(X \mid S)$ factorizes as follows,[2]

$$P_\pi \left( \bar{\boldsymbol{x}}_{1:T}, \bar{\boldsymbol{s}}_{1:T}, \bar{\boldsymbol{y}}_{1:T} \right) = P(s_1) \prod_{t=1}^{T} \pi(x_t \mid s_t) \mathcal{T}(s_t, x_t, s_{t+1}) \mathcal{R}(s_t, x_t, y_t), \tag{1}$$

where the transition distribution $\mathcal{T}$ and the reward distribution $\mathcal{R}$ are *interventional* queries given by

$$\mathcal{T}(s_t, x_t, s_{t+1}) = P_{x_t}(s_{t+1} \mid s_t) = \int_{\boldsymbol{u}_t} \mathbb{1} \left\{ s_{t+1} = f_S(s_t, x_t, \boldsymbol{u}_t) \right\} P(\boldsymbol{u}_t) \tag{2}$$

$$\mathcal{R}(s_t, x_t, y_t) = P_{x_t}(y_t \mid s_t) = \int_{\boldsymbol{u}_t} \mathbb{1} \left\{ y = f_Y(s_t, x_t, \boldsymbol{u}_t) \right\} P(\boldsymbol{u}_t) \tag{3}$$

---

[2]The decomposition holds since state $S_t$ blocks all backdoor paths from action $X_t$ to nodes $S_{t+1}$ and $Y_t$, i.e., path starting with arrow $X_t \leftarrow S_t$. It follows from Rule 2 of do-calculus [32, Theorem 3.4.1] that $P_\pi(y_t \mid s_t, x_t) = P_{x_t}(s_{t+1} \mid s_t)$ and $P_\pi(y_t \mid s_t, x_t) = P_{x_t}(y_t \mid s_t)$.

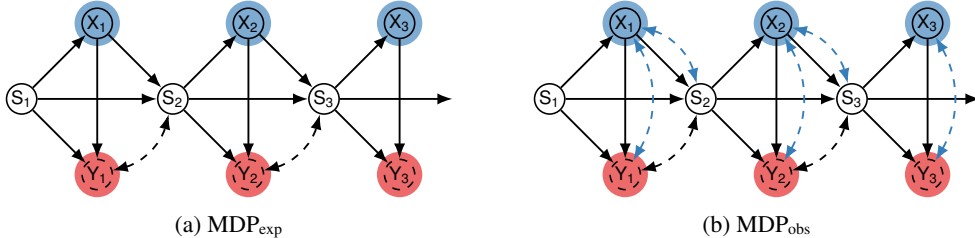

(a) MDP$_{\text{exp}}$                     (b) MDP$_{\text{obs}}$

Figure 2: Causal diagrams where $S_t$ represents the state, $X_t$ represents the action (shaded blue) and $Y_t$ represents the latent reward (shaded red). (a) MDP$_{\text{exp}}$ describes the imitator's interaction with the environment; (b) MDP$_{\text{obs}}$ shows the data-generating process for the expert demonstrations.

For analytical clarity, we define reward function $\mathcal{R}(s, x)$ as the expected value $\sum_y y\mathcal{R}(s, x, y)$. Fix the discounted factor $\gamma \in [0, 1]$. A common objective for an agent to optimize is the cumulative return $R_t = Y_t + \gamma Y_{t+1} + \gamma^2 Y_{t+2} + \cdots = \sum_{k=0}^{\infty} \gamma^k Y_{t+k}$.

**Imitation Learning.** When a detailed parametrization of the transition distribution $\mathcal{T}$ and the reward function $\mathcal{R}$ is available, the agent can obtain an optimal policy using standard planning algorithms [36, 6]. However, in many practical applications, complete knowledge of these parametrizations is often unavailable, necessitating a learning process. In this paper, we consider the *imitation learning* setting, where the agent has access to observed trajectories generated by the expert. More specifically, at each time step $t$, the expert selects an action $X_t \leftarrow f_X(s_t, \boldsymbol{u}_t)$ based on the current state $S_t = s_t$ and latent noise $\boldsymbol{U}_t = \boldsymbol{u}_t$. Fig. 2b shows the graphical representation of the data-generating process of the expert; the highlighted bi-directed arrows, e.g., $X_t \leftrightarrow Y_t$, indicate the presence of an unobserved confounder $U \in \boldsymbol{U}_t$ affecting both the action $X_t$ and outcome $Y_t$. We summarize the expert trajectories using the observational distribution $P(\boldsymbol{X}, \boldsymbol{S}, \boldsymbol{Y})$ over sequences of variables $\boldsymbol{X} = \{X_1, X_2, \dots\}$, $\boldsymbol{S} = \{S_1, S_2, \dots\}$, and $\boldsymbol{Y} = \{Y_1, Y_2, \dots\}$. It is verifiable from Fig. 2b that Markov property holds with regard to distribution $P(\boldsymbol{X}, \boldsymbol{S}, \boldsymbol{Y})$. For any horizon $T$,

$$P\left(\bar{\boldsymbol{x}}_{1:T}, \bar{\boldsymbol{s}}_{1:T}, \bar{\boldsymbol{y}}_{1:T}\right) = P(s_1) \prod_{t=1}^{T} P(x_t \mid s_t) \widetilde{\mathcal{T}}(s_t, x_t, s_{t+1}) \widetilde{\mathcal{R}}(s_t, x_t, y_t) \tag{4}$$

where $\widetilde{\mathcal{T}}$ and $\widetilde{\mathcal{R}}$ are the expert's nominal transition distribution and reward function computed from the *observational* distribution as follows:

$$\widetilde{\mathcal{T}}(s, x, s') = P\left(S_{t+1} = s' \mid S_t = s, X_t = x\right), \qquad \widetilde{\mathcal{R}}(s, x) = \mathbb{E}\left[Y_t \mid S_t = s, X_t = x\right] \tag{5}$$

By convention in imitation learning, we assume the rewards $Y_t$ are generally unobserved to the learner; instead, it has access to a parametric family $\mathscr{R}$ containing the expert's nominal reward function $\mathbb{E}\left[Y_t \mid s_t, x_t\right]$. Given the expert demonstrations $\mathcal{D}$ sampled from $P(X_1, X_2, \dots, S_1, S_2, \dots)$ and the parametric reward family $\mathscr{R}$, the imitator attempts to learn policy $\pi$ that can achieve expert performance, i.e., $\mathbb{E}_\pi\left[\sum_{t=1}^{\infty} \gamma^{t-1} Y_t\right] \geq \mathbb{E}\left[\sum_{t=1}^{\infty} \gamma^{t-1} Y_t\right]$. Standard imitation methods focus on the identifiable setting where the imitator's transition distribution $\mathcal{T}$ and reward function $\mathcal{R}$ is consistent with the expert's nominal transition $\widetilde{\mathcal{T}}$ and reward $\widetilde{\mathcal{R}}$. Formally,

**Definition 2** (Causal Consistency). For an interventional distribution $P_\pi$ and an observational distribution $P$ satisfying the Markov property (Def. 1), Causal Consistency is said to hold with respect to $P_\pi$ and $P$ if the following statement is true, for every time step $t = 1, 2, \dots$,

$$P_{x_t}\left(s_{t+1} \mid s_t\right) = P\left(s_{t+1} \mid s_t, x_t\right), \quad \text{and} \quad P_{x_t}\left(y_t \mid s_t\right) = P\left(y_t \mid s_t, x_t\right) \tag{6}$$

When the invariances of Def. 2 hold, the learner could recover the parametrization of the transition distribution $\mathcal{T}$ from observational data $P(\boldsymbol{X}, \boldsymbol{S})$ and infer about the reward function $\mathcal{R}$ from the parametric family $\mathscr{R}$. An imitating policy $\pi$ is obtainable by solving the following minimax program,

$$\nu^* = \min_{\boldsymbol{\pi}} \max_{\mathcal{R} \in \mathscr{R}} \sum_{s, x} \mathcal{R}(s, x) \left(P(x \mid s)\rho(s) - \pi(x \mid s)\rho_\pi(s)\right) \tag{7}$$

where the imitator's $\rho_\pi$ and the expert's $\rho$ occupancy measures are defined as, respectively, $\rho_\pi(s) = \sum_{t=0}^{\infty} \gamma^t P_\pi\left(S_t = s\right)$ and $\rho(s) = \sum_{t=0}^{\infty} \gamma^t P\left(S_t = s\right)$. The solution $\pi$ is guaranteed to achieve expert

performance when the gap $\nu^* \leq 0$. This means that the imitator following policy $\pi$ performs as well as the expert, even in the worst-case environment instance compatible with the demonstration data and model assumption. Several imitation learning algorithms have been proposed to solve the optimization problem in Eq. (7), including [1, 50, 19].

Graphical criteria exist [33, 46, 34] to examine whether causal consistency (Def. 2) holds from causal knowledge of the environment, including the celebrated *backdoor* condition [32, Def. 3.3.1],[15]. In MDPs, this means that the causal links between the latent noise $U_t$ and action $X_t$ are not effective - the graphical representation of the imitator's (Fig. 2a) and the expert's (Fig. 2b) data-generating process coincide. However, in practice, causal consistency could be fragile and does not necessarily hold due to the presence of unobserved confounders in the demonstration data [57, 39]. The remainder of this paper studies imitation learning when violations occur in the invariance relationships of Eq. (6).

## 2.1 Imitation with Non-Identifiable Transition and Reward

We first consider the imitation setting described in Fig. 2b where unobserved confounders generally exist in the expert demonstrations; both the transition distribution $\mathcal{T}$ and reward function $\mathcal{R}$ are not identifiable from Eq. (6). Here, we will show that expert performance is not imitable by constructing worst-case MDP instances where the expert always outperforms the imitator.

The state value function $V_\pi(s)$ is defined as the expected return given the imitator's starting state $S_t = s$ following a policy $\pi$, i.e., $V_\pi(s) = \mathbb{E}_\pi[R_t \mid S_t = s]$. For any policy $\pi$, the imitator's performance can be written as $\mathbb{E}_\pi[R_1] = \sum_{s_1} P(s_1)V_\pi(s_1)$. The value function of any state $s$ can thus be recursively defined using the celebrated *Bellman Equation* [6]:

$$V_\pi(s) = \sum_x \pi(x \mid s) \left( \mathcal{R}(s,x) + \gamma \sum_{s'} \mathcal{T}(s,x,s')V_\pi(s') \right) \tag{8}$$

where $\gamma$ denotes the discount factor. While the transition distribution $\mathcal{T}$ and the reward function $\mathcal{R}$ are not uniquely discernible from the observational distribution due to the unobserved confounding, it is still possible to learn about them from demonstrations using partial identification. Without loss of generality, the reward $Y_t$ is normalized in a real interval $[0, 1]$. Through rigorous adaptation of the bounding strategies established in [29, 55], we successfully derive the bounds for the transition distribution $\mathcal{T}$ and reward function $\mathcal{R}$, for every realization $(s, x, s') \in \mathcal{S} \times \mathcal{X} \times \mathcal{S}$,

$$\mathcal{T}(s,x,s') \in \left[ \widetilde{\mathcal{T}}(s,x,s') P(x \mid s), \ \widetilde{\mathcal{T}}(s,x,s') P(x \mid s) + P(\neg x \mid s) \right] \tag{9}$$

$$\mathcal{R}(s,x) \in \left[ \widetilde{\mathcal{R}}(s,x) P(x \mid s), \ \widetilde{\mathcal{R}}(s,x) P(x \mid s) + P(\neg x \mid s) \right] \tag{10}$$

Among the above quantities, $\widetilde{\mathcal{T}}$ and $\widetilde{\mathcal{R}}$ are the expert's nominal transition distribution and reward function in Eq. (5); $P(x \mid s)$ stands for the propensity score $P(X_t = x \mid S_t = s)$ and $P(\neg x \mid s) = 1 - P(x \mid s)$. We can then construct a worst-case MDP for any policy $\pi$ at state $s$ by solving the following optimization program: minimize the Bellman's equation in Eq. (8) as the objective function, subject to the observational constraints in Eqs. (9) and (10). Solving this program enables a valid MDP construction since the transition distribution $\mathcal{T}$ and the reward function $\mathcal{R}$ are independent components induced by the underlying model and can be optimized separately.

**Theorem 1.** *Given any positive observational distribution $P(\boldsymbol{X}, \boldsymbol{S}, \boldsymbol{Y}) > 0$, there exists an MDP model $\hat{M}$ compatible with the causal graph of Fig. 2b such that $P(\boldsymbol{X}, \boldsymbol{S}, \boldsymbol{Y}; \hat{M}) = P(\boldsymbol{X}, \boldsymbol{S}, \boldsymbol{Y})$ and for any policy $\pi$, any time step $t = 1, 2, \ldots$, any state $s \in \mathcal{S}$,*

$$V_\pi\left(s; \hat{M}\right) < \mathbb{E}\left[R_t \mid S_t = s; \hat{M}\right]. \tag{11}$$

In other words, there always exists a candidate MDP instance $\hat{M}$ compatible with the demonstration data such that an imitator is always unable to achieve expert performance (r.h.s. in Eq. (11)), regardless of the deployed policy $\pi$. It follows from Thm. 1 that there is no policy $\pi$ learnable from confounded demonstrations that is guaranteed to perform at least as the expert in all possible scenarios. This means that expert performance is not imitable when unobserved confounding generally exists. The following example demonstrates the challenges of unobserved confounding in a single-stage MDP.

**Example 1** (Single-Stage MDP)**.** Consider a 1-stage MDP model with horizon $T = 1$. For any policy $\pi(X_1 \mid S_1)$, the imitator's expected return is $\mathbb{E}_\pi[Y_1] = \sum_{s_1,x_1} \mathcal{R}(s_1, x_1)\pi(x_1 \mid s_1)P(s_1)$. It follows

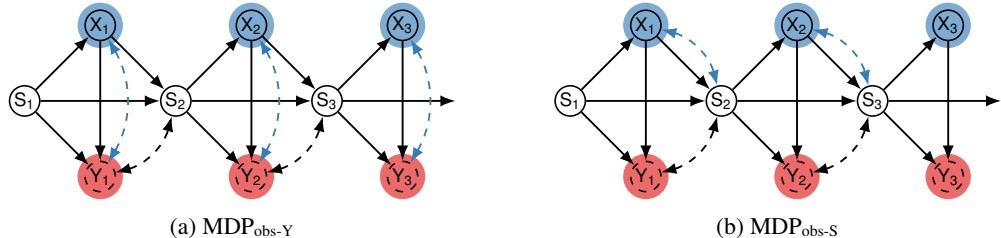

<p style="text-align:center">(a) MDP_obs-Y          (b) MDP_obs-S</p>

Figure 3: (a) $\text{MDP}_{\text{obs-Y}}$ shows a data-generating process for expert demonstrations where only the reward $Y_t$ is confounded with the action $X_t$; (b) $\text{MDP}_{\text{obs-S}}$ shows a data-generating process for expert demonstrations where only the next state $S_{t+1}$ is confounded with the action $X_t$.

from the tight lower bound in Eq. (10) that there exists an worst-case MDP model $\hat{M}$ compatible with the observational distribution $P(X_1, S_1, Y_1)$ such that $\mathcal{R}(s_1, x_1) = \mathbb{E}[Y_1 \mid s_1, x_1] P(x_1 \mid s_1)$. In this MDP instance $\hat{M}$, the imitator's expected return can be further written as

$$\mathbb{E}_\pi[Y_1] = \sum_{s_1, x_1} \mathbb{E}[Y_1 \mid s_1, x_1] P(x_1|s_1) \pi(x_1|s_1) P(s_1) < \sum_{s_1, x_1} \mathbb{E}[Y_1 \mid s_1, x_1] P(x_1|s_1) P(s_1) \quad (12)$$

The last step holds since probabilities of the policy $\pi(x_1 \mid s_1) \in [0, 1]$ and $\sum_{x_1} \pi(x_1 \mid s_1) = 1$. Marginalizing the above equation gives $\mathbb{E}_\pi[Y_1] < \mathbb{E}[Y_1]$ - the imitator is unable to achieve expert performance regardless of the deployed policy $\pi$. This analysis applies analogously to the MAB model in Fig. 1, which can be thought of as a 1-stage MDP with no initial state $S_1 = \emptyset$. We refer the readers to Appendix F for more examples about the 2-stage MDP.

## 3 Partial Identification for Robust Imitation

The impossibility results in Thm. 1 imply that robust imitation cannot be guaranteed when unobserved confounders generally exist in the demonstration data. This means we must explore alternative assumptions to learn an imitating policy guaranteed to achieve expert performance. Meanwhile, standard imitation methods apply when causal consistency of Def. 2 holds, and no unobserved confounder affects the transition or reward function. A natural question at this point arises: whether robust imitation is feasible for settings between the unconfounded (Fig. 1b) and fully confounded cases (Fig. 1a), where unobserved confounding bias affects only either the transition distribution or reward function? This section aims to answer this question.

### 3.1 Imitation with Identifiable Transition and Non-Identifiable Reward

We first examine the setting graphically described in Fig. 3a where the reward function is confounded, while the transition distribution is identifiable from the demonstration data. In this case, the first equation of Def. 2 holds while the second one fails. To initiate the discussion, we write the expected return of a candidate policy $\pi$ in an MDP environment as follows [36],

$$\mathbb{E}_\pi[R_1] = \sum_{s,x} \mathcal{R}(s, x) \pi(x \mid s) \rho_\pi(s) \quad (13)$$

Among quantities in the above equation, the state occupancy measure $\rho_\pi(s) = \sum_{t=0}^\infty \gamma^t P_\pi(S_t = s)$ is a function of the initial state distribution $P(s) = P(S_1 = s)$ and the transition distribution $\mathcal{T}$. Specifically, $\rho_\pi(s)$ can be recursively written as $\rho_\pi(s) = P(s) + \gamma \sum_{s', x} \mathcal{T}(s', x, s) \pi(x \mid s') \rho_\pi(s')$. When the transition distribution is unconfounded (Fig. 3a), one could recover its parametrization $\mathcal{T}(s, x, s')$ following the first formula of Def. 1. Therefore, what remains undetermined in Eq. (13) is the non-identifiable reward function $\mathcal{R}$. It follows from Eq. (10) that parametrization of $\mathcal{R}(s, x)$ can be bounded from the observational distribution. The imitator's expected return could thus be lower bounded as $\mathbb{E}_\pi[R_1] \geq \sum_{s,x} \widetilde{\mathcal{R}}(s, x) P(x \mid s) \pi(x \mid s) \rho_\pi(s)$, where $\widetilde{\mathcal{R}}$ is the nominal reward function defined in Eq. (5). Similarly, the expert's expected return could be decomposed as $\mathbb{E}[R_1] = \sum_{x,s} \widetilde{\mathcal{R}}(s, x) P(x \mid s) \rho(s)$, where $\rho(s) = \sum_{t=0}^\infty \gamma^t P(S_t = s)$ is the expert's occupancy measure. Optimizing the worst-case gap between the imitator $\mathbb{E}_\pi[R_1]$ and expert $\mathbb{E}[R_1]$ leads to a minimax optimization problem, the solution of which leads to a possible imitating policy.

<p style="text-align:center">6</p>

---

**Algorithm 1:** Causal GAIL with Confounded Reward $\mathcal{R}$ (CAIL-$\mathcal{R}$)

---

1: **Input**: Expert demonstrations $\mathcal{D} = \{(S_i, X_i)\}_{i=1}^N$
2: **for** iteration $k = 0, 1, 2, \ldots$ **do**
3:    Collect expert trajectories from $\mathcal{D}$
4:    Collect imitator trajectories based on the policy $\pi_k(x \mid s)$
5:    Update the parameters $w$ of discriminator $D_k$ with gradient

$$\hat{\mathbb{E}}[\nabla_w \log(D_k(s,x))] + \hat{\mathbb{E}}_{\pi_k}[\nabla_w P(x \mid s) \log(1 - D_k(s,x))] \tag{15}$$

6:    Update the policy $\pi_{k+1} = \arg\min_\pi \mathbb{E}_\pi[P(x \mid s) \log(1 - D(s,x))]$ using any forward RL algorithm
7: **end for**

---

---

**Algorithm 2:** Causal GAIL with Confounded Transition $\mathcal{T}$ (CAIL-$\mathcal{T}$)

---

1: **Input**: Expert demonstrations $\mathcal{D} = \{(S_i, X_i)\}_{i=1}^N$
2: **for** iteration $k = 0, 1, 2, \ldots$ **do**
3:    Collect expert trajectories from $\mathcal{D}$
4:    Collect imitator trajectories based on the policy $\pi_k(x \mid s)$ from the worst-case occupancy measure by solving the optimization problem presented in Eq. (19) and Eq. (20)
5:    Update the parameters $w$ of discriminator $D_k$ with gradient

$$\hat{\mathbb{E}}[\nabla_w \log(D_k(s,x))] + \hat{\mathbb{E}}_{\pi_k}[\nabla_w \log(1 - D_k(s,x)); \mathcal{T}] \tag{16}$$

6:    Update the policy $\pi_{k+1} = \arg\min_\pi \mathbb{E}_\pi[\log(1 - D(s,x)); \mathcal{T}]$ with any forward RL algorithm
7: **end for**

---

**Theorem 2.** *Given an MDP $M$ compatible with the causal graph of Fig. 3a, let $\mathcal{R}$ be a parametric family containing the conditional reward $\mathbb{E}[Y_t \mid s_t, x_t]$. Consider the following optimization program,*

$$\nu^* = \min_\pi \max_{\widetilde{\mathcal{R}} \in \mathcal{R}} \sum_{s,x} \widetilde{\mathcal{R}}(s,x) P(x \mid s) \left(\rho(s) - \pi(x \mid s)\rho_{\boldsymbol{\pi}}(s)\right) \tag{14}$$

*When the gap $\nu^* \leq 0$, the solution $\pi^*$ is an imitating policy satisfying $\mathbb{E}_{\pi^*}[R_1] \geq \mathbb{E}[R_1]$.*

In other words, Thm. 2 computes an imitating policy within the environment depicted in Fig. 3a by finding a policy maximizing the worst-case reward function compatible with the demonstration data and the expert's nominal reward. Later in Sec. 4, we will demonstrate that such a solution exists and robust imitation learning is feasible in Fig. 3a.

The optimization program in Thm. 2 could be solved by augmenting some standard imitation learning such as GAIL [19]. To make the argument more precise, let the parametric family $\mathcal{R}$ be a set of reward function $\mathcal{R}(s,x)$ taking values in the real space $\mathbb{R}$. We penalize the complexity of a reward function $\mathcal{R}$ by subtracting a convex regularization function $\psi(\mathcal{R})$ from Eq. (14); the detailed definition of $\psi(\mathcal{R})$ is given by [19, Eq. 13]. Solving the optimization program of Eq. (14) is equivalent to matching weighted occupancy measures between the imitator and the expert, shown in Appendix B,

$$\nu^* = \min_\pi \psi^* \left(P(x \mid s)\rho(s) - P(x \mid s)\pi(x \mid s)\rho_\pi(s)\right) \tag{17}$$

$$= \min_\pi \max_{D \in (0,1)^{\mathcal{S} \times \mathcal{X}}} \mathbb{E}[\log(D(S,X))] + \mathbb{E}_\pi\left[P(x \mid s)\log(1 - D(S,X))\right], \tag{18}$$

where $\psi^* = \max_\mathcal{R} a^\top \mathcal{R} - \psi(\mathcal{R})$ is a conjugate function of $\psi$; function $D \in \mathcal{S} \times \mathcal{X} \mapsto (0,1)$ is a discriminator classifier (e.g, a neural network). The above optimization problem is in the form of two neural networks competing against each other in a zero-sum game. The detailed implementation of our proposed algorithm, called CAIL-$\mathcal{R}$, is provided in Alg. 1. Compared to the standard GAIL algorithm, Alg. 1 adds weight to the signal generated by the discriminator for the imitator and then attempts to match the distribution between the weighted samples and expert demonstrations.

## 3.2 Imitation with Non-Identifiable Transition and Identifiable Reward

In this section, we examine the $\text{MDP}_{\text{obs-S}}$ environment as graphically depicted in Fig. 3b, where the reward function is unconfounded, but UCs affect the action $X_t$ and the next state $S_{t+1}$ simultaneously. In this setting, the second equation of Causal Consistency (Def. 2) is satisfied, aligning the reward function $\mathcal{R}$ with the expert's nominal reward function. However, the first equation of Def. 2 does not generally hold due to confounding bias, making the transition distribution $\mathcal{T}$ not identifiable from demonstrations. Despite these challenges, we utilize partial identification techniques to bound the transition function $\mathcal{T}$, and subsequently estimate the imitator's performance.

More precisely, consider again the expected return decomposition in Eq. (13). The identifiable reward function $\mathcal{R}$ must be contained in the parametric space of the expert's nominal reward $\mathscr{R}$. The transition distribution $\mathcal{T}$ can be bounded from the demonstration data using Eq. (9). One could thus obtain a lower bound over the imitator's performance by reasoning about the worst-case occupancy measure compatible with demonstrations. Formally, with the fixed reward function $\mathcal{R}$ and the fixed policy $\pi$, the imitator's return $\mathbb{E}_\pi[R_1]$ is bounded by:

$$\mathbb{E}_\pi[R_1] \geq \min_{\mathcal{T}, \rho_\pi} \quad \sum_{s,x} \mathcal{R}(s,x)\pi(x \mid s)\rho_\pi(s) \tag{19}$$

$$\text{s.t.: } \rho_\pi(s) \geq 0, \quad \sum_s \rho_\pi(s) = \frac{1}{1-\gamma}, \quad \text{and } \rho_\pi(s) = P(s) + \gamma \sum_{s',x} \mathcal{T}(s',x,s)\pi(x \mid s')\rho_\pi(s')$$

$$\text{Obs. Constraints } \mathscr{T}: \quad \begin{cases} \sum_s \mathcal{T}(s',x,s) = 1, \quad \text{and } \mathcal{T}(s,x,s') \geq \widetilde{\mathcal{T}}(s,x,s')P(x \mid s) \\ \mathcal{T}(s,x,s') \leq \widetilde{\mathcal{T}}(s,x,s')P(x \mid s) + P(\neg x \mid s) \end{cases} \tag{20}$$

The above optimization problem is similar to the classical linear program for planning in MDPs [36]. The main difference is that the transition distribution $\mathcal{T}$ is no longer fixed but bounded in a convex space $\mathscr{T}$ specified from the observational data. Therefore, we develop an imitating policy by minimizing the performance gap between the imitator and the expert in the worst-case environment compatible with the observational data and prior knowledge.

**Theorem 3.** *Given an MDP $M$ compatible with the causal graph of Fig. 3b, let $\mathscr{R}$ be a parametric family containing the conditional reward $\mathbb{E}[Y_t \mid s_t, x_t]$, and $\mathscr{T}$ be a parametric family over conditional probabilities $P(s_{t+1} \mid s_t, x_t)$ defined in Eq. (20). Consider the following program,*

$$\nu^* = \min_\pi \max_{\mathcal{R} \in \mathscr{R}} \max_{\mathcal{T} \in \mathscr{T}} \sum_{s,x} \mathcal{R}(s,x)\left(P(x \mid s)\rho(s) - \pi(x \mid s)\rho_{\boldsymbol{\pi}}(s; \mathcal{T})\right) \tag{21}$$

*When the gap $\nu^* \leq 0$, the solution $\pi^*$ is an imitating policy satisfying $\mathbb{E}_{\pi^*}[R_1] \geq \mathbb{E}[R_1]$.*

We solve the optimization program in Thm. 3 by augmenting GAIL, a standard imitation method [19]. By penalizing the complexity of a reward function $\mathcal{R}$ using a convex regularization function $\psi(\mathcal{R})$ from Eq. (14), Eq. (21) is reducible to the following distribution matching problem,

$$\nu^* = \min_\pi \max_{D \in (0,1)^{\mathcal{S} \times \mathcal{X}}} \max_{\mathcal{T} \in \mathscr{T}} \mathbb{E}[\log(D(S,X))] + \mathbb{E}_\pi[\log(1 - D(S,X)); \mathcal{T}], \tag{22}$$

We present the step-by-step implementation of our imitation method, CAIL-$\mathcal{T}$, in Alg. 2. It is similar to the standard GAIL [19]; however, a significant distinction arises at step 4, where the imitator collects trajectories from the worst-case occupancy measure as presented in Eq. (19) and Eq. (20), which is obtainable by iteratively solving a series of linear programs. We refer readers to Appendix C for a more detailed discussion, where we propose an iterative algorithm designed to find the worst-case occupancy measure efficiently.

# 4 Experiments

In this section, we validate the theoretical findings presented in Thm. 1 and illustrate the applications of the proposed CAIL algorithms (Alg. 1 and Alg. 2) on various causal imitation learning tasks. Such tasks range from synthetic causal models to real-world scenarios. To summarize, when both the transition and the reward are confounded, there always exists a worst-case MDP instance $\hat{M}$ compatible with the expert demonstrations, but the imitator consistently fails to match expert

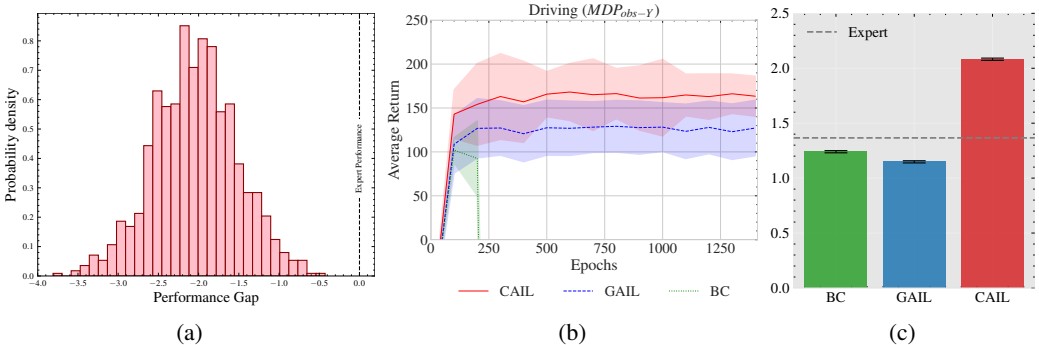

(a)  (b)  (c)

Figure 4: Simulation results for our experiments. Fig. 4a illustrates the performance gap histogram for the experiment MDP$_{obs}$, where negative values indicate performance worse than expert performance. Fig. 4b shows the convergence plot for CAIL, GAIL, and BC performance. Fig. 4c shows the final performance, where y-axis represents the expected return.

performance, aligning with the proof provided in Sec. 2.1. When either the transition or the reward is confounded, we systematically evaluate our algorithms against the standard BC and GAIL methods, highlighting the importance of optimizing within the worst-case SCM. Standard BC mimics the expert's nominal behavior policy $P(X|S)$ via supervised learning; standard GAIL learns a policy by solving a min-max game [19]. We provide in Appendix D more details on the experiment setup.

**MDP$_{obs}$ – Random Instances.** This experiment aims to empirically validate the theoretical findings discussed in Thm. 1. Consider SCM instances compatible with Fig. 2b including binary observed variables $S_t, X_t, Y_t \in \{0, 1\}$. 1000 random discrete MDPs are sampled, in other words, the reward functions and the transition probabilities are generally different among these models. The expert is able to observe the state $S_t$, the unobserved variable $U_t$. However, the imitator, lacking access to both $U_t$ or the reward $\mathbb{E}_\pi[Y_t]$, makes decisions solely on $S_t$. As shown in Fig. 4a, imitators consistently failed to match expert performance. Specifically, prevalent negative performance gaps indicate that most of imitators were consistently worse than experts; only in rare cases did the performance gaps near $-0.5$, supporting our theoretical insights in Thm. 1. In summary, imitators fail to achieve the expert's performance when both the reward and the transition are confounded.

**MDP$_{obs-Y}$ – Driving.** To demonstrate the proposed framework, as outlined in Alg. 1, we consider a scenario when an autonomous vehicle ('ego vehicle') aims to learn optimal driving strategies from expert demonstrations. The state $S_t$ contains some critical driving information, e.g., the velocities of the ego vehicle and the leading vehicle and the spatial distance between them. The action $X_t$ represents acceleration or deceleration decisions the ego vehicle makes. The unobserved variable $U_t$ represents some information accessible to the expert but inaccessible to the imitator, e.g. slippery road conditions [26]. The reward $Y_t$ is designed to reflect multiple realistic driving objectives, e.g., safety, comfort, efficiency, and so on. $U_t$ has an effect on the reward $Y_t$. Unlike the scenarios described in [39, 42], due to UCs between $X_t$ and $Y_t$ at each step $t$, it is impossible to find a $\pi$-backdoor admissible set. BC, GAIL, and CAIL utilize the same policy space $\pi(x \mid s)$. The major difference between CAIL and GAIL lies in that CAIL optimizes the imitator by the weighted reward generated from the discriminator $- P(x \mid s) \log(1 - D(s, x))$. As illustrated in Fig. 4b, where means and standard deviations are computed over 100 trajectories, CAIL consistently outperforms BC and GAIL.

**MDP$_{obs-S}$ – Medical Treatment.** Consider the challenge of providing medical treatment to acutely ill patients, where the primary goal is to learn a policy so that the morality rate can be decreased. We utilize the real-world medical treatment dataset, i.e., Medical Information Mart for Intensive Care III (MIMIC-III) dataset [22]. MIMIC-III consists trajectories of clinical information (e.g., heart rate, oxygen saturation, and so on) recorded at various time intervals. However, due to privacy concerns, certain essential variables are masked or not properly recorded [45], e.g., socioeconomic status or the experience levels of caregivers [9, 56]. Specifically, the state $S_t$ encapsulates the critical health information for the patients, e.g., prolonged elevated heart rate (peHR). The action $X_t$ represents whether to treat the medicine or not. The reward $Y_t$ is designed to represent the intent of the doctor as much as possible, e.g., avoiding the patient's mortality. The unobserved confounded

$U_t$ simultaneously affects the action $X_t$ and the next state $S_{t+1}$. Simulation results are illustrated in Fig. 4c, which shows that the proposed framework performs the best among all strategies. BC and IRL fail to obtain an imitating policy that could match expert performance.

## 5 Conclusion

This paper investigates imitation learning in Markov Decision Processes where the unobserved confounding bias cannot be ruled out *a priori*. We establish theoretically that when such unobserved confounders generally exist, it is infeasible to obtain a robust imitating policy that can perform at least as well as the expert across all possible environments compatible with the demonstration data and prior knowledge. Departing from this critical realization, our research diverges into two distinct problem settings – one where only the transition distribution is unconfounded, but the reward function is non-identifiable due to unobserved confounding; and the other where the reward function is unconfounded and the transition distribution is non-identifiable. We then propose novel imitation learning algorithms using partial identification techniques, which allow the imitator to obtain effective policies that can achieve expert performance for both problem settings. Through extensive experiments, we empirically validate the theoretical findings and systematically evaluate our algorithms on different scenarios, ranging from simulated causal models to real-world datasets.

## Acknowledgements

This research was supported in part by the NSF, ONR, AFOSR, DoE, Amazon, JP Morgan, and The Alfred P. Sloan Foundation.

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

# A    Proofs

In this section, we provide proofs for the theoretical claims delineated in the paper. Throughout this paper, it is important to note that detailed parametrizations of the underlying SCM are not known to the agent. Instead, the agent has access to the expert's demonstrations, which are summarized as the observational distribution $P(\boldsymbol{X}, \boldsymbol{S}, \boldsymbol{Y})$.

We begin by revisiting the distribution of state visitation. Specifically, $\rho_\pi(s)$ can be calculated by:

$$\rho_\pi(s) = P(s) + \gamma \sum_{s',x} \mathcal{T}(s', x, s)\pi(x \mid s')\rho_\pi(s') \tag{23}$$

where $P(s)$ represents the initial state distribution, $\gamma$ represents the discount factor, $\mathcal{T}$ represents the transition probabilities for the imitator. Subsequently, we are able to develop the occupancy measure for the policy $\pi$:

$$\rho_\pi(s, x) = \rho_\pi(s)\pi(x \mid s) \tag{24}$$

It is important to note that, although the format of the occupancy measure $\rho_\pi(s, x)$ shares a formal resemblance to the one presented in GAIL [19], $\rho_\pi(s, x)$ specifically represents an interventional distribution with policy $\text{do}(\pi)$. The identifiability of the transition $\mathcal{T}(s, x, s')$ directly impacts the identifiability of $P_\pi(s_t)$. If $P_\pi(s_t)$ is not identifiable, $\rho_\pi(s)$ and $\rho_\pi(s, x)$ are consequently not identifiable.

**Theorem 1.** *Given any positive observational distribution $P(\boldsymbol{X}, \boldsymbol{S}, \boldsymbol{Y}) > 0$, there exists an MDP model $\hat{M}$ compatible with the causal graph of Fig. 2b such that $P(\boldsymbol{X}, \boldsymbol{S}, \boldsymbol{Y}; \hat{M}) = P(\boldsymbol{X}, \boldsymbol{S}, \boldsymbol{Y})$ and for any policy $\pi$, any time step $t = 1, 2, \ldots$, any state $s \in \mathcal{S}$,*

$$V_\pi\left(s; \hat{M}\right) < \mathbb{E}\left[R_t \mid S_t = s; \hat{M}\right]. \tag{11}$$

*Proof.* Without loss of generality, the reward $Y$ is normalized so that it has a range of $[0, 1]$ Based on the value function defined in Eq. (8), we first show how to expand it into a recursive version:

$$V_\pi(s_t) = \mathbb{E}_\pi\left[\sum_{k=0}^\infty \gamma^k Y_{t+k} \mid s_t\right] \tag{25}$$

$$= \mathbb{E}_\pi[Y_t \mid s_t] + \mathbb{E}_\pi\left[\sum_{k=1}^\infty \gamma^k Y_{t+k} \mid s_t\right] \tag{26}$$

$$= \mathbb{E}_\pi[Y_t \mid s_t] + \gamma\mathbb{E}_\pi\left[\sum_{k=1}^\infty \gamma^{k-1} Y_{t+k} \mid s_t\right] \tag{27}$$

$$= \mathbb{E}_\pi[Y_t \mid s_t] + \gamma \sum_{s_{t+1}} P_\pi(s_{t+1} \mid s_t)\mathbb{E}_\pi\left[\sum_{k=0}^\infty \gamma^k Y_{t+1+k} \mid s_t, s_{t+1}\right] \tag{28}$$

$$= \mathbb{E}_\pi[Y_t \mid s_t] + \gamma \sum_{s_{t+1}} P_\pi(s_{t+1} \mid s_t)V_\pi(s_{t+1}), \tag{29}$$

where $\gamma$ is the discount factor, $P_\pi(s_{t+1} \mid s_t)$ denotes the transition probability when executing policy $\pi$.

From the second last line to the last line is justified by the experimental markovian property, as discussed in Sec. 2, following the graph Fig. 2a. More details could be found in [54]. $\mathbb{E}_\pi[Y_t \mid s_t] = \mathbb{E}[Y_t \mid s_t, \text{do}(\pi)]$ denotes the expected reward received by the agent when executing policy $\pi$. Similarly, the transition probability

$$P_\pi(s_{t+1} \mid s_t) = \sum_{x_t} P_{x_t}(s_{t+1} \mid s_t)\pi(x_t \mid s_t), \tag{30}$$

and $P_{x_t}(s_{t+1} \mid s_t) = P(s_{t+1} \mid s_t, \text{do}(x_t)) = \mathcal{T}(s_t, x_t, s_{t+1})$. Generally speaking, when any unobserved confounder exists between $S_{t+1}$ and $X_t$, the causal query $P_{x_t}(s_{t+1} \mid s_t)$ is not identifiable

[32, 43, 5, 51]. Building on the previous derivations, we arrive at the recursive formulation of the value function under policy $\pi$:

$$V_\pi(s_t) = \sum_{x_t} \mathcal{R}(s_t, x_t)\,\pi(x_t \mid s_t) + \gamma \sum_{s_{t+1}} \mathcal{T}(s_t, x_t, s_{t+1})\pi(x_t \mid s_t)V_\pi(s_{t+1}) \tag{31}$$

$$= \sum_{x_t} \pi(x_t \mid s_t)\left(\mathcal{R}(s_t, x_t) + \gamma \sum_{s_{t+1}} \mathcal{T}(s_t, x_t, s_{t+1})V_\pi(s_{t+1})\right). \tag{32}$$

Next, to establish the validity of the preceding claim, we proceed by applying the technique of mathematical induction. Let $|S|$ denote the number of distinct states for $S$.

**Base case $t = T$.** For the final timestep $T$, for each state index $j$ where $\forall j, 1 \le j \le |S|$, the value function $V_\pi(s_{(T,j)})$ cane be defined as follows:

$$\begin{aligned}V_\pi(s_{(T,j)}) &= \mathbb{E}_\pi\left[Y_T \mid S_T = s_{(T,j)}\right]\\ &= \sum_{x_t} \mathbb{E}_{x_t}\left[Y_T \mid S_T = s_{(T,j)}\right]\pi(x_t \mid s_{(T,j)})\end{aligned} \tag{33}$$

where $s_{(T,j)}$ refers to the scenario where the state at the final timestep $S_T$ is equal the specific state $j$.

In order to obtain the worst-case SCM $\hat{M}$, we need to minimize $V_\pi(s_T) - V(s_T)$ compatible with the observational distribution, by establishing its lower bound. To this end, we directly employ the natural bound [29], which has been discussed in Sec. 2.1:

$$\begin{aligned}\min_M \quad & V_\pi(s_{(T,j)}; M) - V(s_{(T,j)}; M)\\ &= \sum_{x_t} \mathbb{E}_{x_t}\left[Y_T \mid S_T = s_{(T,j)}; M\right]\pi(x_t \mid s_{(T,j)}) - V(s_{(T,j)}; M)\\ &= \sum_{x_t} \mathbb{E}\left[Y_T \mid s_{(T,j)}, x_t\right]P(x_t \mid s_{(T,j)})\pi(X_T = x_t \mid s_{(T,j)}) - \sum_{x_t} \mathbb{E}\left[Y_T \mid s_{(T,j)}, x_t\right]P(x_t \mid s_{(T,j)})\\ &< 0\end{aligned}$$

$$\tag{34}$$

The last step is justified because $P(\boldsymbol{X}, \boldsymbol{S}, \boldsymbol{Y}) > 0$ and $0 \le \pi(X_T = x_t \mid s_{(T,j)}) \le 1$. Intuitive examples illustrating this conclusion are provided in Sec. 2.1 and Appendix F. Therefore, this confirms the validity of the inequality for the base case.

Specifically, in certain degenerate cases where there is only one possible action, the imitator has no choice but to follow that single option. Consequently, unobserved confounders are less likely to introduce significant effects in these scenarios. However, under such conditions, pursuing imitation learning is not meaningful, as there is no variability in choice for the imitator to learn from. Therefore, such cases are of limited relevance to the scope of this analysis.

**Induction case.** Suppose at $t + 1$, $V_\pi(s_{t+1}) < V(s_{t+1})$, we need to prove $V_\pi(s_t) < V(s_t)$.

$$V_\pi(s_t) = \mathbb{E}_\pi[Y_t \mid s_t] + \gamma \sum_j P_\pi(s_{(t+1,j)} \mid s_t)\underbrace{V_\pi(s_{(t+1,j)})}_{<V(s_{(t+1,j)})} \tag{35}$$

Without loss of generality, we assume that the state with the minimal value at $t + 1$ is denoted as $s_{(t+1,|S|)}$. Our approach is founded on the premise that in obtaining the worst-case SCM $\hat{M}$, it is strategic to allocate the lowest possible transition probabilities to the state with the highest value, while preferentially assigning higher probabilities to states demonstrating smaller values. Specifically, one starts with the estimate $P(S_{t+1} = s_{(t+1,1)}, x_t \mid S_t)$ for $P_{x_t}(S_{t+1} = s_{(t+1,1)} \mid S_t)$. Following this logic, we systematically allocate probability masses for indices $1 \le j \le |S| - 1$ as follows:

$$P_{x_t}(S_{t+1} = s_{(t+1,j)} \mid S_t) \leftarrow P(S_{t+1} = s_{(t+1,j)}, x_t \mid S_t)$$

In accordance with the established properties of probability distributions, it follows that:

$$\sum_{j=1}^{|S|} P_{x_t}(S_{t+1} = s_{(t+1,j)} \mid S_t) = 1.$$

Considering the state $s_{(t+1,|S|)}$, the corresponding probability can be assigned as:

$$P_{x_t}(S_{t+1} = s_{(t+1,|S|)} \mid S_t) = 1 - \sum_{j=1}^{|S|-1} P_{x_t}(S_{t+1} = s_{(t+1,j)} \mid S_t).$$

By substituting the assigned values, we are able to derive the following expression:

$$P_{x_t}(S_{t+1} = s_{(t+1,|S|)} \mid S_t) \leftarrow 1 - P(s_{(t+1,1)}, x_t \mid S_t) - P(s_{(t+1,2)}, x_t \mid S_t) \cdots - P(s_{(t+1,|S|-1)}, x_t \mid S_t),$$

where the right-hand side simplifies to:

$$\left(P(s_{(t+1,1)}, x_t \mid S_t) + P(s_{(t+1,2)}, x_t \mid S_t) \cdots + P(s_{(t+1,|S|-1)}, x_t \mid S_t)\right) = \left(\sum_{j=1}^{|S|-1} P\left(s_{(t+1,j)}, x_t \mid S_t\right)\right)$$

$$= P(x_t \mid S_t) - P(s_{(t+1,|S|)}, x_t \mid S_t).$$

It is established that the expression $0 \leq 1 - P(x_t \mid S_t) + P(s_{(t+1,|S|)}, x_t \mid S_t) \leq 1$ holds true. This inequality is supported by the following equation:

$$\sum_{j=1}^{|S|} P(s_{(t+1,j)}, x_t \mid S_t) = P(x_t \mid S_t).$$

To further analyze the expert policy, the associated value function $V(s_t)$ can be expanded as follows:

$$V(s_t) = \mathbb{E}\left[\sum_{k=0}^{\infty} \gamma^k Y_{t+k} \mid S_t = s_t\right]$$

$$= \mathbb{E}[Y_t \mid s_t] + \gamma \sum_j P(s_{(t+1,j)} \mid s_t) V(s_{(t+1,j)}),$$

(36)

where $\gamma$ is the discount factor, $P(s_{(t+1,j)} \mid s_t)$ denotes the observational transition probability. Notably, $P(s_{(t+1,j)} \mid s_t)$ and $P_\pi(s_{(t+1,j)} \mid s_t)$ are generally different, because they reflect two distinct probabilities: $P(s_{(t+1,j)} \mid s_t)$ corresponding to the observational distribution and the other, $P_\pi(s_{(t+1,j)} \mid s_t)$, representing the imitator's transition dynamics.

In accordance with the established properties of probability distributions, it follows that:

$$\sum_{j=1}^{|S|} P(S_{t+1} = s_{(t+1,j)} \mid S_t) = 1.$$

Without loss of generality, suppose the policy is a deterministic policy. Actually, the following proof holds true regardless of the choice of $x_t$. Subsequently, we analyze the gap between $V_\pi(s_t)$ and $V(s_t)$ as follows:

$$V_\pi(s_t) - V(s_t)$$

$$= \left(\mathbb{E}_\pi[Y_t \mid s_t] + \gamma \sum_{j=1}^{|S|} P_\pi(s_{(t+1,j)} \mid s_t) V_\pi(s_{(t+1,j)})\right)$$

$$- \left(\mathbb{E}[Y_t \mid s_t] + \gamma \sum_{j=1}^{|S|} P(s_{(t+1,j)} \mid s_t) V(s_{(t+1,j)})\right)$$

(37)

$$= \left(\mathbb{E}_\pi[Y_t \mid s_t] + \gamma \sum_{j=1}^{|S|} \sum_{x_t} P_{x_t}(s_{(t+1,j)} \mid s_t) \pi(x_t \mid s_t) V_\pi(s_{(t+1,j)})\right)$$

$$- \left(\mathbb{E}[Y_t \mid s_t] + \gamma \sum_{j=1}^{|S|} P(s_{(t+1,j)} \mid s_t) V(s_{(t+1,j)})\right)$$

$$\min_M \quad V_\pi(s_t; M) - V(s_t; M)$$

$$= \mathbb{E}_\pi[Y_t \mid s_t; M] - \mathbb{E}[Y_t \mid s_t; M] + \gamma \sum_{j=1}^{|S|-1} \left( P(s_{(t+1,j)}, x_t \mid s_t) - P(s_{(t+1,j)} \mid s_t) \right) V(s_{(t+1,j)})$$

$$+ \gamma \left( 1 - \left( \sum_{j=1}^{|S|-1} P\left(s_{(t+1,j)}, x_t \mid s_t\right) \right) - P(s_{(t+1,|S|)} \mid s_t) \right) V(s_{(t+1,|S|)})$$

$$= \underbrace{\mathbb{E}_\pi[Y_t \mid s_t; M] - \mathbb{E}[Y_t \mid s_t; M]}_{<0}$$

$$+ \gamma \sum_{j=1}^{|S|-1} \underbrace{\left( P(s_{(t+1,j)}, x_t \mid s_t) - P(s_{(t+1,j)} \mid s_t) \right)}_{<0} \underbrace{\left( V(s_{(t+1,j)}) - V(s_{(t+1,|S|)}) \right)}_{>0}$$

$$< 0$$

(38)

where $\min_M \mathbb{E}_\pi[Y_t \mid s_t; M] - \mathbb{E}[Y_t \mid s_t; M] < 0$ follows a similar logic as previously introduced in the base case, and $V(s_{(t+1,j)}) - V(s_{(t+1,|S|)}) > 0$ is consistent with the ordering assumption, where the state $s_{(t+1,|S|)}$ represents the minimal value.

In some degenerated cases when $\mathbb{E}_\pi[Y_t \mid s_t] = 0$ and $\mathbb{E}[Y_t \mid s_t] = 0$, it might coincidentally follow that $V_\pi(s_t) = 0$, which is equal to $V(s_t) = 0$. Another instance of degeneracy occurs when the value function $V(s_{(t+1,j)})$ remains the same across all states. Such occurrences are extremely unlikely in practical scenarios, especially when $P(\boldsymbol{X}, \boldsymbol{S}, \boldsymbol{Y}) > 0$. $\qquad \square$

## B   Derivations for Causal GAIL Algorithms

**Theorem 2.** *Given an MDP $M$ compatible with the causal graph of Fig. 3a, let $\mathscr{R}$ be a parametric family containing the conditional reward $\mathbb{E}[Y_t \mid s_t, x_t]$. Consider the following optimization program,*

$$\nu^* = \min_\pi \max_{\widetilde{\mathcal{R}} \in \mathscr{R}} \sum_{s,x} \widetilde{\mathcal{R}}(s, x) P(x \mid s) \left( \rho(s) - \pi(x \mid s) \rho_{\boldsymbol{\pi}}(s) \right) \tag{14}$$

*When the gap $\nu^* \leq 0$, the solution $\pi^*$ is an imitating policy satisfying $\mathbb{E}_{\pi^*}[R_1] \geq \mathbb{E}[R_1]$.*

*Proof.* Based on Eq. (13), we have:

$$\mathbb{E}_\pi[R_1] = \sum_{s,x} \mathcal{R}(s, x) \pi(x \mid s) \rho_\pi(s)$$

It follows from Eq. (10) that parametrization of $\mathcal{R}(s, x)$ can be bound from the observational distribution. The imitator's expected return could thus be lower bounded as

$$\mathbb{E}_\pi[R_1] \geq \sum_{s,x} \widetilde{\mathcal{R}}(s, x) P(x \mid s) \pi(x \mid s) \rho_\pi(s)$$

Note that the expert's expected return could be similarly decomposed as

$$\mathbb{E}[R_1] = \sum_{x,s} \widetilde{\mathcal{R}}(s, x) P(x \mid s) \rho(s),$$

where $\rho(s) = \sum_{t=0}^\infty \gamma^t P(S_t = s)$ is the expert's occupancy measure.

$$\nu^* = \min_\pi \max_M \quad \mathbb{E}[R_1; M] - \mathbb{E}_\pi[R_1; M] \tag{39}$$

$$= \min_\pi \max_{\widetilde{\mathcal{R}}, \mathcal{R}} \quad \sum_{x,s} \widetilde{\mathcal{R}}(s, x) P(x \mid s) \rho(s) - \sum_{s,x} \mathcal{R}(s, x) \pi(x \mid s) \rho_\pi(s) \tag{40}$$

$$= \min_\pi \max_{\widetilde{\mathcal{R}}} \quad \sum_{s,x} \widetilde{\mathcal{R}}(s, x) P(x \mid s) \left( \rho(s) - \pi(x \mid s) \rho_{\boldsymbol{\pi}}(s) \right), \tag{41}$$

which is the ultimate target expression. $\qquad \square$

Next, we will show the derivation details for matching weighted occupancy measures between the imitator and the expert. Suppose $\psi^* = \max_{\mathcal{R}} a^\top \mathcal{R} - \psi(\mathcal{R})$ is a conjugate function of $\psi$. Following a similar logic in [19], we utilize a smiliar cost regularizer $\psi_{GA}$, leading to the formulation of Alg. 1. Basically, Alg. 1 minimizes Jensen-Shannon divergence between $P(x \mid s)\rho(s)$ and $P(x \mid s)\pi(x \mid s)\rho_\pi(s)$.

First, we reformulate the equation into state-action occupancy measures:

$$\psi^* \left( P(x \mid s)\rho(s) - P(x \mid s)\pi(x \mid s)\rho_\pi(s) \right) = \psi^* \left( \rho(s,x) - P(x \mid s)\rho_\pi(s,x) \right) \tag{42}$$

Based on the definition of $\psi^*$, we have:

$$\psi^* \left( \rho(s,x) - P(x \mid s)\rho_\pi(s,x) \right) \tag{43}$$

$$= \max_{\mathcal{R}} \sum_{s,x} \left( \rho(s,x) - P(x \mid s)\rho_\pi(s,x) \right) \mathcal{R}(s,x) - \sum_{s,x} P(x \mid s)\rho_\pi(s,x) g_\phi(\mathcal{R}(s,x)) \tag{44}$$

$$= \sum_{s,x} \max_{\mathcal{R}} \rho(s,x)\mathcal{R} - P(x \mid s)\rho_\pi(s,x)\phi \left( -\phi^{-1}(-\mathcal{R}) \right) \tag{45}$$

$$= \sum_{s,x} \max_{\mathcal{R}'} \rho(s,x)(-\phi(\mathcal{R}')) - P(x \mid s)\rho_\pi(s,x)\phi \left( -\phi^{-1}(\phi(\mathcal{R}')) \right) \tag{46}$$

$$= \sum_{s,x} \max_{\mathcal{R}'} \rho(s,x)(-\phi(\mathcal{R}')) - P(x \mid s)\rho_\pi(s,x)\phi \left( -\mathcal{R}' \right) \tag{47}$$

where we make the change of variables $\mathcal{R} \to -\phi(\mathcal{R}')$. Suppose $D \in \mathcal{S} \times \mathcal{X} \mapsto (0,1)$ is a discriminator classifier (e.g, a neural network). Using the logistic loss $\phi(x) = \log \left( 1 + e^{-x} \right)$, we can get:

$$\psi^* \left( \rho(s,x) - P(x \mid s)\rho_\pi(s,x) \right) \tag{48}$$

$$= \sum_{s,x} \max_{\mathcal{R}'} \rho(s,x) \log \left( \frac{1}{1 + e^{-\mathcal{R}'}} \right) + P(x \mid s)\rho_\pi(s,x) \log \left( 1 - \frac{1}{1 + e^{-\mathcal{R}'}} \right) \tag{49}$$

$$= \max_{D \in (0,1)^{\mathcal{S} \times \mathcal{X}}} \mathbb{E}[\log(D(S,X))] + \mathbb{E}_\pi \left[ P(x \mid s) \log(1 - D(S,X)) \right], \tag{50}$$

which is the ultimate target expression.

**Theorem 3.** *Given an MDP $M$ compatible with the causal graph of Fig. 3b, let $\mathscr{R}$ be a parametric family containing the conditional reward $\mathbb{E}[Y_t \mid s_t, x_t]$, and $\mathscr{T}$ be a parametric family over conditional probabilities $P \left( s_{t+1} \mid s_t, x_t \right)$ defined in Eq. (20). Consider the following program,*

$$\nu^* = \min_\pi \ \max_{\mathcal{R} \in \mathscr{R}} \ \max_{\mathcal{T} \in \mathscr{T}} \ \sum_{s,x} \mathcal{R}(s,x) \left( P(x \mid s)\rho(s) - \pi(x \mid s)\rho_{\boldsymbol{\pi}} \left( s; \mathcal{T} \right) \right) \tag{21}$$

*When the gap $\nu^* \leq 0$, the solution $\pi^*$ is an imitating policy satisfying $\mathbb{E}_{\pi^*} [R_1] \geq \mathbb{E}[R_1]$.*

*Proof.* Based on Eq. (13), we have:

$$\mathbb{E}_\pi [R_1] = \sum_{s,x} \mathcal{R} (s,x) \pi(x \mid s)\rho_\pi(s)$$

$$= \sum_{s,x} \mathcal{R} (s,x) \underbrace{\rho_\pi(s,x)}_{\text{Non-ID}}$$

The reward function $\mathcal{R}$ is identifiable and must be contained in the parametric space of the expert's nominal reward $\mathscr{R}$. In other words,

$$\mathcal{R} (s,x) = \widetilde{\mathcal{R}} (s,x). \tag{51}$$

The transition distribution $\mathcal{T}$ can be bounded from the demonstration data using Eq. (9). Therefore, we get:

$$\nu^* = \min_\pi \max_M \quad \mathbb{E}[R_1; M] - \mathbb{E}_\pi[R_1; M] \tag{52}$$

$$= \min_\pi \max_{\mathcal{T},\widetilde{\mathcal{R}},\mathcal{R}} \quad \sum_{s,x} \widetilde{\mathcal{R}}(s,x)P(x \mid s)\rho(s) - \mathcal{R}(s,x)\rho_\pi(s,x;\mathcal{T}) \tag{53}$$

$$= \min_\pi \max_{\mathcal{T},\mathcal{R}} \quad \sum_{s,x} \mathcal{R}(s,x)\left(P(x \mid s)\rho(s) - \rho_\pi(s,x;\mathcal{T})\right) \tag{54}$$

$$= \min_\pi \max_{\mathcal{T}\in\mathscr{T},\mathcal{R}\in\mathscr{R}} \quad \sum_{s,x} \mathcal{R}(s,x)\left(P(x \mid s)\rho(s) - \pi(x \mid s)\rho_\pi(s;\mathcal{T})\right) \tag{55}$$

which is the ultimate desired expression. $\qquad\square$

Consider again the expected return decomposition in Eq. (13). The reward function $\mathcal{R}$ is identifiable and must be contained in the parametric space of the expert's nominal reward $\mathscr{R}$. The transition distribution $\mathcal{T}$ can be bounded from the demonstration data using Eq. (9). One could thus obtain a lower bound over the imitator's performance by reasoning about the worst-case occupancy measure compatible with demonstrations. Formally, with the fixed reward function $\mathcal{R}$ and the fixed policy $\pi$, the imitator's return is bounded by

$$\mathbb{E}_\pi[R_1] \geq \min_{\mathcal{T},\rho_\pi} \quad \sum_{s,x} \mathcal{R}(s,x)\pi(x \mid s)\rho_\pi(s) \tag{56}$$

$$\text{subject to:} \quad \rho_\pi(s) \geq 0, \quad \text{and} \sum_s \rho_\pi(s) = \frac{1}{1-\gamma}$$

$$\rho_\pi(s) = P(s) + \gamma \sum_{s',x} \mathcal{T}(s',x,s)\,\pi(x \mid s')\rho_\pi(s')$$

$$\text{Obs. Constraints } \mathscr{T}: \quad \begin{cases} \sum_s \mathcal{T}(s',x,s) = 1, \quad \text{and } \mathcal{T}(s,x,s') \geq \widetilde{\mathcal{T}}(s,x,s')\,P(x \mid s) \\ \mathcal{T}(s,x,s') \leq \widetilde{\mathcal{T}}(s,x,s')\,P(x \mid s) + P(\neg x \mid s) \end{cases} \tag{57}$$

The above optimization problem is similar to the classic linear program for planning in MDPs [36]. The main difference is that the transition distribution $\mathcal{T}$ is no longer fixed but bounded in a convex space $\mathscr{T}$ specified from the observational data. Similar to the previous setting, we could solve an imitating policy by minimizing the performance gap between the imitator and the expert in the worst-case environment compatible with the observational data and prior knowledge.

Next, we will provide a heuristic algorithm to solve the optimization program presented in Eq. (19) and Eq. (20). Specifically, as discussed in Eq. (9), we are able to bound the transition distribution $\mathcal{T}$ by:

$$\mathcal{T}(s,x,s') \in \left[\widetilde{\mathcal{T}}(s,x,s')\,P(x \mid s), \widetilde{\mathcal{T}}(s,x,s')\,P(x \mid s) + P(\neg x \mid s)\right]. \tag{58}$$

The intuition for Alg. 3 is: in order to find the worst case, we need to put as less transition probability mass as possible to the state with maximal values, and allocate higher transition probabilities to states with smaller values. Without loss of generality, suppose $V_{x_t}(s_{(t+1,|S|)})$ is found to have the smallest relative value. For all other states $j \neq |S|$, we need to allocate as less transition probability mass as possible. Therefore, we take the lower bound:

$$P_{x_t}(S_{t+1} = s_{(t+1,j)} \mid s_t) := P(S_{t+1} = s_{(t+1,j)}, x_t \mid s_t) \tag{59}$$

$$:= P(S_{t+1} = s_{(t+1,j)} \mid s_t, x_t)P(x_t \mid s_t), \tag{60}$$

where $P_{x_t}(s_{t+1} \mid s_t) = P(s_{t+1} \mid s_t, \mathrm{do}(x_t)) = \mathcal{T}(s_t, x_t, s_{t+1})$, and $P(s_{(t+1)} \mid s_t, x_t) = \widetilde{\mathcal{T}}(s_t, x_t, s_{t+1})$. For the state $s_{(t+1,|S|)}$, we have:

$$P_{x_t}(S_{t+1} = s_{(t+1,|S|)} \mid s_t) := 1 - \left(\sum_{j=1}^{|S|-1} P(S_{t+1} = s_{(t+1,j)}, x_t \mid s_t)\right). \tag{61}$$

**Algorithm 3:** Find Worst-Case Discounted Future Reward

---

1: **Input**: $P(s_{t+1}, x_t|s_t)$, the value function $V_{x_t}(s_t)$
2: **Output**: Probability mass assignments for non-ID transitions $\mathcal{T}$
3: Let $V_{x_t}(s_{(t+1,|S|)})$ is determined to have the minimal relative value
4: Set

$$P_{x_t}(S_{t+1} = s_{(t+1,j)} \mid s_t) := P(S_{t+1} = s_{(t+1,j)}, x_t \mid s_t), \text{where } j \neq |S|$$

$$P_{x_t}(S_{t+1} = s_{(t+1,|S|)} \mid s_t) := 1 - \left( \sum_{j=1}^{|S|-1} P(S_{t+1} = s_{(t+1,j)}, x_t \mid s_t) \right)$$

5: **return**

---

Following a similar logic in Alg. 1: we reformulate the equation into state-action occupancy measures:

$$\psi^* \left( P(x \mid s)\rho(s) - \pi(x \mid s)\rho_\pi(s; \mathcal{T}) \right) = \psi^* \left( \rho(s, x) - \rho_\pi(s, x; \mathcal{T}) \right) \tag{62}$$

Based on the definition of $\psi^*$, we have:

$$\psi^* \left( \rho(s, x) - \rho_\pi(s, x; \mathcal{T}) \right) \tag{63}$$

$$= \max_{\mathcal{T}, \mathcal{R}} \sum_{s,x} \left( \rho(s, x) - \rho_\pi(s, x; \mathcal{T}) \right) \mathcal{R}(s, x) - \sum_{s,x} \rho_\pi(s, x; \mathcal{T}) g_\phi(\mathcal{R}(s, x)) \tag{64}$$

$$= \sum_{s,x} \max_{\mathcal{T}, \mathcal{R}} \rho(s, x)\mathcal{R} - \rho_\pi(s, x; \mathcal{T})\phi \left( -\phi^{-1}(-\mathcal{R}) \right) \tag{65}$$

$$= \sum_{s,x} \max_{\mathcal{T}, \mathcal{R}'} \rho(s, x)(-\phi(\mathcal{R}')) - \rho_\pi(s, x; \mathcal{T})\phi \left( -\phi^{-1}(\phi(\mathcal{R}')) \right) \tag{66}$$

$$= \sum_{s,x} \max_{\mathcal{T}, \mathcal{R}'} \rho(s, x)(-\phi(\mathcal{R}')) - \rho_\pi(s, x; \mathcal{T})\phi \left( -\mathcal{R}' \right) \tag{67}$$

Suppose $D \in \mathcal{S} \times \mathcal{X} \mapsto (0, 1)$ is a discriminator classifier (e.g, a neural network). Using the logistic loss $\phi(x) = \log \left( 1 + e^{-x} \right)$, we can get:

$$\psi^* \left( \rho(s, x) - \rho_\pi(s, x; \mathcal{T}) \right) \tag{68}$$

$$= \sum_{s,x} \max_{\mathcal{T}, \mathcal{R}'} \rho(s, x) \log \left( \frac{1}{1 + e^{-\mathcal{R}'}} \right) + \rho_\pi(s, x; \mathcal{T}) \log \left( 1 - \frac{1}{1 + e^{-\mathcal{R}'}} \right) \tag{69}$$

$$= \max_{\mathcal{T}, D} \mathbb{E}[\log(D(S, X))] + \mathbb{E}_\pi \left[ \log(1 - D(S, X)); \mathcal{T} \right]. \tag{70}$$

Therefore, we are able to obtain the ultimate target expression:

$$\nu^* = \min_\pi \max_{\mathcal{T} \in \mathscr{T}, D \in (0,1)^{\mathcal{S} \times \mathcal{X}}} \mathbb{E}[\log(D(S, X))] + \mathbb{E}_\pi \left[ \log(1 - D(S, X)); \mathcal{T} \right]. \tag{71}$$

## C  Finding the Worst-Case Transition Distribution

In this section, we provide a practical algorithm, Alg. 3, designed to solve the optimization problem formulated in Eq. (19) and Eq. (20). The underlying rationale of Alg. 3 is to search for the worst-case scenario by allocating the minimal transition probability mass to the state with the highest value while assigning greater transition probabilities to states with lower values. The resulting solution should still adhere to a set of predefined observational constraints to ensure feasibility. This approach ensures that the most "adversarial" outcome is prioritized during the optimization process.

To further clarify the approach above, consider the following numerical example. Suppose there are only two states. The value function $V_{x_t}(s_{t+1})$ takes on two values: $V_{x_t}(s_{(t+1,1)}) = 0.8$ and $V_{x_t}(s_{(t+1,2)}) = 0.2$. Because $V_{x_t}(s_{(t+1,1)}) > V_{x_t}(s_{(t+1,2)})$, the algorithm seeks the worst-case discounted future reward by allocating $P_{x_t}(s_{(t+1,1)} \mid s_t) \leftarrow P(s_{(t+1,1)}, x_t \mid s_t)$ and $P_{x_t}(s_{(t+1,2)} \mid s_t) \leftarrow 1 - P(s_{(t+1,1)}, x_t \mid s_t)^3$. As such, we are able to collect trajectories from the imitator, even though $P_\pi(s_{t+1} \mid s_t)$ is not identifiable.

---

[3]In this case, although $P_{x_t}(s_{(t+1,2)} \mid s_t)$ has a lower bound of $P(s_{(t+1,2)}, x_t \mid s_t)$, it cannot be set exactly equal to this value. It is crucial to maintain the condition $P_{x_t}(s_{(t+1,1)} \mid s_t) + P_{x_t}(s_{(t+1,2)} \mid s_t) = 1$.

# D   More Details for the Experiments

All experiments were conducted using Intel Cascade Lake processors, with 30 vCPUs and 120 GB memory on a system running Ubuntu 18.04. Upon acceptance of this manuscript, we intend to make the source code available in the camera-ready version of the paper.

**MDP$_\text{obs}$**   Previously, 1000 random discrete causal models are sampled and all the performance gaps are less than 0. In other words, when both the reward and the transition are confounded, all imitators fail to match expert performance.

Specifically, let's take a look at one example instance of those randomly sampled SCM instances. Its detailed parameterization is provided as follows:

$$
\begin{aligned}
P(s_0) &= 0.5, & P(s_1) &= 0.5 \\
P(x_0, y_0, s_0' \mid s_0) &= 0.1888, & P(x_0, y_0, s_1' \mid s_0) &= 0.2099, \\
P(x_0, y_1, s_0' \mid s_0) &= 0.0294, & P(x_0, y_1, s_1' \mid s_0) &= 0.2116, \\
P(x_1, y_0, s_0' \mid s_0) &= 0.1465, & P(x_1, y_0, s_1' \mid s_0) &= 0.0226, \\
P(x_1, y_1, s_0' \mid s_0) &= 0.0645, & P(x_1, y_1, s_1' \mid s_0) &= 0.1267, \\
P(x_0, y_0, s_0' \mid s_1) &= 0.1762, & P(x_0, y_0, s_1' \mid s_1) &= 0.1775, \\
P(x_0, y_1, s_0' \mid s_1) &= 0.0290, & P(x_0, y_1, s_1' \mid s_1) &= 0.1786, \\
P(x_1, y_0, s_0' \mid s_1) &= 0.1761, & P(x_1, y_0, s_1' \mid s_1) &= 0.0893, \\
P(x_1, y_1, s_0' \mid s_1) &= 0.1472, & P(x_1, y_1, s_1' \mid s_1) &= 0.0261,
\end{aligned}
\tag{72}
$$

where $s'$ denotes the next state; $P(x_0, y_0, s_0' \mid s_0)$ is the abbreviation format for $P(X_t = x_0, Y_t = y_0, S_{t+1} = s_0' \mid S_t = s_0)$.

The expert is able to observe the state $S_t$, the unobserved variable $U_t$, and the reward $Y_t$. However, the imitator, lacking access to both $U_t$ or the reward $\mathbb{E}_\pi[Y_t]$, makes decisions solely on $S_t$. In other words, all methods utilize the same policy scope $\pi(x \mid s)$. As shown in Fig. 4a, imitators consistently failed to match expert performance. Prevalent negative performance gaps indicate that most of imitators were significantly worse than experts; only in rare cases did the performance gaps near $-0.5$, supporting our theoretical insights presented in Thm. 1. Furthermore, as depicted in Fig. 5a, CAIL does not achieve expert-level performance, specifically, $\mathbb{E}_\pi[R_t] - \mathbb{E}[R_t] = -1.9019$. However, although CAIL performs worse than the expert, CAIL still consistently outperforms BC and GAIL by effectively learning from the constructed worst-case MDP instances.

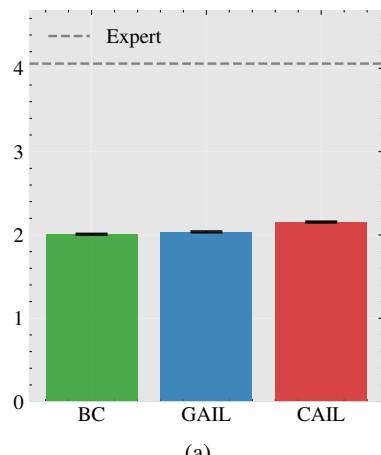
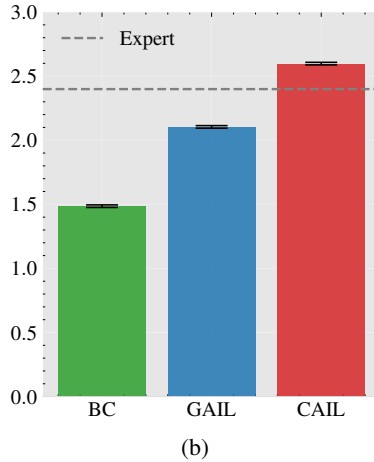

(a)                                                  (b)

Figure 5: Simulation results for experiments that are not included in the main manuscript.

**MDP$_\text{obs-Y}$: Additional Experiment.** Consider an SCM instance compatible with Fig. 3a including binary observed variables $S_t, X_t, Y_t \in \{0, 1\}$. $S_t$ represents the state at each time step. $X_t$ denotes the action. The unobserved variable $U_t$ represents some information accessible to the expert but

inaccessible to the imitator. Additionally, the imitator lacks access to the reward $\mathbb{E}_\pi[Y_t]$. Its detailed parameterization is provided as follows:

$$
\begin{aligned}
& P(s_0) = 0.5, && P(s_1) = 0.5 \\
& P(x_0, y_0, s_0' \mid s_0) = 0.1775, && P(x_0, y_0, s_1' \mid s_0) = 0.2029, \\
& P(x_0, y_1, s_0' \mid s_0) = 0.0001, && P(x_0, y_1, s_1' \mid s_0) = 0.0001, \\
& P(x_1, y_0, s_0' \mid s_0) = 0.0993, && P(x_1, y_0, s_1' \mid s_0) = 0.0199, \\
& P(x_1, y_1, s_0' \mid s_0) = 0.2001, && P(x_1, y_1, s_1' \mid s_0) = 0.3001, \\
& P(x_0, y_0, s_0' \mid s_1) = 0.2859, && P(x_0, y_0, s_1' \mid s_1) = 0.1359, \\
& P(x_0, y_1, s_0' \mid s_1) = 0.0001, && P(x_0, y_1, s_1' \mid s_1) = 0.0001, \\
& P(x_1, y_0, s_0' \mid s_1) = 0.2969, && P(x_1, y_0, s_1' \mid s_1) = 0.2809, \\
& P(x_1, y_1, s_0' \mid s_1) = 0.0001, && P(x_1, y_1, s_1' \mid s_1) = 0.0001,
\end{aligned}
\tag{73}
$$

where $s'$ denotes the next state; $P(x_0, y_0, s_0' \mid s_0)$ is the abbreviation format for $P(X_t = x_0, Y_t = y_0, S_{t+1} = s_0' \mid S_t = s_0)$. As depicted in Fig. 5b, CAIL performs the best among all strategies. Both BC and GAIL fail to match expert performance. Such result shows the effectiveness of Alg. 1.

## E   Broader Impacts

This paper investigates the theoretical framework of causal imitation learning from confounded demonstrations. Our framework is versatile, applicable to various real-world domains such as autonomous driving, robotics, industrial automation, and medical decisions modeling. One of the positive impacts of this study is the exploration of the risks associated with training IRL algorithms when demonstrations are generally contaminated by unobserved confounders. We theoretically prove that when both the transition distribution $\mathcal{T}$ and reward function $\mathcal{R}$ are not identifiable, there is no policy $\pi$ learnable from confounded demonstrations that is guaranteed to perform at least as the expert in all possible scenarios. Such theoretical findings have been validated through extensive randomly generated causal models. When either the reward function or the transition distribution is confounded, we augment the GAIL framework by utilizing partial identification techniques, so that the imitator is optimized within the worst-case scenarios. Specfically, the worst-case reward function in Alg. 1 and the worst-case occupancy measure in Alg. 2. By mitigating the risks associated with unobserved confounders in expert demonstrations, our framework supports the development of more transparent and accountable AI systems. This transparency is crucial in high-stakes areas such as healthcare and transportation, where decision-making errors can have significant repercussions. More broadly, our framework significantly enhances the reliability and safety of autonomous systems in various fields, which prioritize safety and robustness during their decision-making processes. They are increasingly important because black-box AI systems, – whose internal workings remain opaque – become more and more prevalent, and our understandings of their potential implications remain limited.

## F   Impossibility Result in Two-Stage MDPs

In this extension of the MAB model introduced in Sec. 1, we explore a two-stage framework (see Fig. 2b). Our previous discussions demonstrated that in MAB settings affected by unobserved confounders, the expert consistently outperforms the imitator; that is, i.e., $\mathbb{E}_x[Y] < \mathbb{E}[Y]$.

We now extend our analysis to the two-stage MDPs. Specifically, the agent first observes the state $S_1$, selects an action $X_1$, and subsequently, it receives a reward $Y_1$. The process then progresses to the second stage, where the agent transitions to state $S_2$. It chooses an action $X_2$, and then it receives a further reward $Y_2$. A pivotal distinction between this scenario and prior examples lies in the transition probability $P_{\pi_1}(S_2 \mid S_1)$. Therefore, we investigate their cumulative reward:

$$
\mathbb{E}_{\pi_1, \pi_2}[Y_1 + Y_2] \qquad \text{and} \qquad \mathbb{E}[Y_1 + Y_2].
\tag{74}
$$

As a motivating example, we assume that all variables are binary. Our analysis begins by comparing the performance at the final stage, specifically, $\mathbb{E}_{\pi_1, \pi_2}[Y_2]$.

Suppose $f(S_2) = \mathbb{E}[Y_2 \mid S_2, X_2]P(X_2 \mid S_2)$. Without loss of generality, we assume an ordering in the functional values associated with different states: $f(S_2 = 0) > f(S_2 = 1)$. To address the

non-identifiability issue caused by the transition distribution $P_{\pi_1}(S_2 \mid S_1)$, as discussed in Eq. (9), we formulate the worst-case SCM by allocating $f(S_2 = 0)$ with probability mass $P(S_2 = 0, X_1 \mid S_1)$. In other words, we assign the lower bound $P(S_2 = 0, x_1 \mid Z_1)$ to the non-identifiable query $P_{x_1}(S_2 = 0 \mid Z_1)$. As such, we are able to rewrite the expert's rewards as follows:

$$\mathbb{E}[Y_2] = f(S_2 = 0) * P(S_2 = 0, X_1 = 0|Z_1)P(Z_1) \tag{75}$$
$$+ f(S_2 = 0) * P(S_2 = 0, X_1 = 1|Z_1)P(Z_1) \tag{76}$$
$$+ f(S_2 = 1) * P(S_2 = 1, X_1 = 0|Z_1)P(Z_1) \tag{77}$$
$$+ f(S_2 = 1) * P(S_2 = 1, X_1 = 1|Z_1)P(Z_1) \tag{78}$$

and the imitator's reward can be written as

$$\mathbb{E}_{\pi_1,\pi_2}[Y_2] = \pi_1(X_1 = 0|Z_1) \cdot A + \pi_1(X_1 = 1|Z_1) \cdot B \tag{79}$$
$$A = f(S_2 = 0) * P(S_2 = 0, X_1 = 0|Z_1)P(Z_1) \tag{80}$$
$$+ f(S_2 = 1) * (1 - P(S_2 = 0, X_1 = 0|Z_1))P(Z_1) \tag{81}$$
$$B = f(S_2 = 0) * P(S_2 = 0, X_1 = 1|Z_1)P(Z_1) \tag{82}$$
$$+ f(S_2 = 1) * (1 - P(S_2 = 0, X_1 = 1|Z_1))P(Z_1) \tag{83}$$

where is $\mathbb{E}_{\pi_1,\pi_2}[Y_2]$ a convex combination of the quantities $A$ and $B$. Therefore, $\mathbb{E}_{\pi_1,\pi_2}[Y_2] \leq \max\{A, B\}$. Given that $f(S_2 = 0) > f(S_2 = 1)$, we are able to establish that $A < \mathbb{E}[Y_2]$ and $B < \mathbb{E}[Y_2]$. Therefore, $\mathbb{E}_{\pi_1,\pi_2}[Y_2] < \mathbb{E}[Y_2]$. Using a similar rationale introduced in Sec. 1, we get $\mathbb{E}_{\pi_1}[Y_1] < \mathbb{E}[Y_1]$. Consequently,

$$\mathbb{E}_{\pi_1,\pi_2}[Y_1 + Y_2] < \mathbb{E}[Y_1 + Y_2]. \tag{84}$$

In other words, the imitator is unable to learn a policy that can obtain the expert's performance in the worst-case 2-stage MDP compatible with the observational distribution $P(X_1, X_2, S_1, S_2, Y_1, Y_2)$.

