# OpenReview forum: "Causal Imitation for Markov Decision Processes: a Partial Identification Approach"
_NeurIPS.cc/2024/Conference — NeurIPS 2024 poster_

### Official Review · Reviewer_P2hy · 2024-07-11

**Soundness:** 2
**Presentation:** 3
**Contribution:** 2
**Rating:** 4
**Confidence:** 2

**Summary:**

This paper studies causal imitation learning from the perspective of partial identification. First, the authors show a hardness result that when both the transitions and the rewards are confounded, it is not possible to imitate or improve over the expert policy. Going forward, under only reward-confounding or transition-confounding, a new imitation learning objective is proposed using the partial identification bound of the corresponding non-identifiable unknowns, and it is proved that once the objective is optimized under $0$, the learned imitation policy would improve over the expert policy. Experimental result justifies the theoretical findings.

**Strengths:**

**Orginality and Significance:**

The idea of partial identification in causal imitation learning is somehow new. The main results are also quite theoretically sound.

**Quality and Clarity:**

The paper is well written. The conclusions and the findings are clearly presented.

**Weaknesses:**

1. The idea seems like a relatively direct extension of the standard imitation algorithm (e.g., GAIL) with the partial identification method from the causal inference literature.
2. Theoretically, it is still unknown when can we guarantee that the imitator can beat the expert. According to Theorem 2 (resp. Theorem 3), only when the optimization problem (14) (resp. (21)) has solutions with non-positive value would the imitator policy be guaranteed to achieve the same performance or improve over the expert policy. Is it possible to derive hardness results that in the reward-confounding-only case (or the transition-confounding-only case) there still exists instances such that it is impossible to improve over the expert?

**Questions:**

1. The derivation of the partial identification bound, which is central to the method proposed in this work, is missing. I suggest clearly include them in the appendix to make the argument self-content and more convincing, or at least make it clear how the conclusion is made given the existing literature like [27].
2. How to principally specify the (observational) reward function class $\mathcal{R}$​ under the unobserved confounding? That is, since the underlying SCM is not known to the agent a priori, how to equip the learning agent a suitable function class $\mathcal{R}$ to match the observational reward function of the unknown SCM?
3. When the algorithm is applied to a non-confounded MDP (i.e., a standard MDP), how well would the algorithm perform when compared with prior arts designed for non-confounded MDPs? More generally, it seems unknown that how the performance of the proposed algorithms would change w.r.t. the confoundedness of the underlying MDP, e.g., how far the interventional probabilities deviate from the observational probabilities.
4. In Theorem 1, according to the proof, it seems essential to assume that *all* the observational probabilities $P(X,S,Y)$ are positive, which seems too strong an assumption. What if one drop this assumption? Does the conclusion change drastically?
5. Some minor typos:
   - In the last line of the footnote in the end of page 3, the first equation should be $P_{\pi}(s_{t+1}|s_t,x_t) = P_{x_t}(s_{t+1}|s_t)$.
   - The imitator's policy $\pi$ appearing in the subscript of $\rho$ is in a bold font but sometimes not, which is not consistent.

**Limitations:**

Please see the weakness section and the question section above.

---

> ### Author Rebuttal · Authors · 2024-08-07
>
> > _“extension of the standard imitation algorithm (e.g., GAIL) with the partial identification method”_
>
> Firstly, applying partial identification to IL is nontrivial due to the complex interplay between unobserved confounders and the dynamics of MDPs. Partial identification typically deals with static settings in causal inference, but MDPs introduce a temporal dimension where past actions and states can influence future outcomes. Our work extends these methods to handle such temporal dependencies, which involves significant theoretical innovation.
>
> Our proposed algorithms might appear neat and straightforward in their final forms, but this does not mean their derivations are simple. The causal bounding results in Eqs. 9-10 are only applicable to a single time step in MDP. Our contribution significantly extends these bounds across time steps, creating a robust framework that can handle the sequential nature of decision-making in practice, where each decision can influence subsequent states and rewards. We also apply special transformations so that they are tailored to GAIL’s training process. We invite the reviewer to check the Appendix and see if our statement is factual.
>
> ---
> > _“hardness results that in the reward-confounding-only case (or the transition-confounding-only case) there still exists instances such that it is impossible to improve over the expert?”_
>
> Yes, it is possible. Consider the reward-confounding case as an example. One could construct an MDP model which consists of a sequence of independent contextual bandit models with side information $S_1, S_2, \dots, S_t$. For each contextual bandit model, given any context $s_i$, the expected reward of each arm matches the worst-case lower bound $E[Y_i | s_i, x_i]P(s_i|x_i)$. It follows from Example 1 that the imitator in this worst-case model is unable to improve over the expert’s performance.
>
> ---
> > _“The derivation of the partial identification bound ...”_
>
> Thanks for the suggestion. The causal bounds in Eqs. 9-10 follow application of (Manski, 90). More specifically, Manski studies a canonical decision setting where $Z$ represents the context, $X$ is the treatment, and $Y$ is the primary outcome. Manski showed that for any observational distribution $P(X, Y, Z)$, the treatment distribution of intervening on $X$ is bounded by $P(y, x|z) \leq P_x(y|z) \leq P(y, x|z) + P(\neg x |z)$.
>
> To derive Eqs. 9-10, one could focus on the MDP within one timestep $t$. Applying Manski’s bound by setting $S_t$ as the context, $X_t$ as the treatment, and $S_{t+1}$ (or $Y_t$) as the primary outcome leads to the statement. We will include additional discussion in the updated manuscript.
>
> ---
> > _“How to principally specify the (observational) reward function class $\mathcal{R}$ under the unobserved confounding? That is, ..., how to equip the learning agent a suitable function class $\mathcal{R}$ to match the observational reward function of the unknown SCM?”_
>
> We do not require the learner to specify the reward function class $\mathcal{R}$ over the interventional reward $E_x[Y|s]$, but only the observational quantity $E[Y|s, x]$. The observed domain of the state-action pair $(s, x)$ is well-specified and can be determined from the demonstration data. In this case, any measurable observational reward $E[Y|s, x]$ can be approximated using some non-parametric function approximators, e.g., neural networks (NNs). The learner could start with a parametric family of NNs, and further restrict it to inject domain knowledge.
>
> ---
> > _“When applied to a non-confounded MDP (i.e., a standard MDP), how well would the algorithm perform when compared with prior arts designed for non-confounded MDPs? ... How far the interventional probabilities deviate from the observational probabilities.”_
>
> For a standard MDP with no unobserved confounding, our algorithm could obtain a policy that is less effective compared to the one learned by the prior arts. The performance gap depends on how close the causal bounds are to the ground-truth interventional probabilities. However, we view this as a feature of our methods. When there is no unobserved confounding in the environment, we recommend the imitator to follow standard imitation learning procedures. When the imitator cannot exclude the unobserved confounding prior, our method obtains an imitating policy with performance guarantee. Our algorithms learn a policy by searching over the worst-case model. We believe this risk-averse approach is principled when facing uncertainty in offline imitation learning, since other models could deviate significantly from the ground-truth, leading to significantly inferior performance.
>
> ---
> > _“In Theorem 1, ... $P(X, S, Y)$ are positive, ... What if one drop this assumption? Does the conclusion change drastically?”_
>
> We require the probabilities $P(X, S, Y)$ to be positive to exclude degenerated cases that the imitator and the expert performance happen to match. As mentioned in L518-520, "In some degenerated cases when $\mathbb{E}\_{\pi}[Y_{t} \mid s\_{t}] = 0$ and $\mathbb{E}[Y\_{t} \mid s\_{t}] = 0$, it might coincidentally follow that $V\_{\pi}(s_{t}) = 0$, which is equal to $V(s\_{t}) = 0$.”
>
> However, such occurrences are highly impossible in practical scenarios, and are statistically insignificant (measure zero) when one generates MDP instances uniformly at random. Moreover, from an algorithmic perspective, focusing on degenerate cases would divert focus from more prevalent and practically significant scenarios.  As the experiment shown within Fig.4 (a), we have randomly generated over $1000$ MDP instance, and in all cases, the minimal performance gap between the imitator and the expert is negative, i.e., the imitator is unable to achieve the expert’s performance. This empirical evidence further supports the assertion that such degenerate cases, while theoretically possible, are exceedingly rare and do not affect the general applicability and robustness of our proposed methods.

---

> > ### Comment · Reviewer_P2hy · 2024-08-13
> >
> > Thank you very much for your detailed responses!  However, I still remain concerned with the soundness of the proposed algorithm. For example, while I agree with the author that the reward function class is to approximate the observed rewards, I am concerned with how the unobserved hidden (unknown) causal structure would affect the performance. More broadly, it is still unknown when can we guarantee that the imitator given by the algorithm can beat the expert, theoretically. This is less discussed even under a suitable theoretical assumption. Given such concerns, I would maintain my score and still recommend rejection of this paper. I suggest further refinement of the work for publication.

---

> > > ### Author Response · Authors · 2024-08-13
> > >
> > > > _"More broadly, it is still unknown when can we guarantee that the imitator given by the algorithm can beat the expert, theoretically."_
> > >
> > > This condition depends on the quality of the demonstration data and the prior knowledge of the latent reward. For example, when there is no unobserved confounding, one could show that the imitator is guaranteed to perform at least as well as the expert. We acknowledge that deriving closed-form solutions for the improvement condition is an exciting problem. However, we would like to note that our proposed algorithms do provide a numerical condition for policy improvement, similar to the standard inverse RL methods like GAIL. When our proposed algorithm returns the game $\nu^* < 0$, the learned policy is robust and guaranteed to outperform the expert. One could find such instances in Figure 4 (b-c).

---

> ### Author Response · Authors · 2024-08-07
>
> Thanks for taking the time to read the paper and your reviews. We recognize the importance of integrating a theoretical framework based on causality into IRL, especially when either the transition dynamics, the reward function, or both, are confounded. Below, we provide further clarification on our contributions, theoretical contributions and the experimental design to enhance understanding and address your concerns effectively.

---

### Official Review · Reviewer_S4Sm · 2024-07-11

**Soundness:** 3
**Presentation:** 3
**Contribution:** 2
**Rating:** 4
**Confidence:** 3

**Summary:**

The paper addresses challenges in imitation learning when the learner and expert have mismatched sensory capabilities and demonstrations are contaminated with unobserved confounding bias. The authors propose robust imitation learning within the framework of Markov Decision Processes (MDPs) using partial identification. They demonstrate that in the presence of unobserved confounders, learning a policy that guarantees expert performance is generally infeasible. The paper introduces two novel algorithms for imitation learning in partially identifiable settings—when either the transition distribution or the reward function is non-identifiable. These algorithms, based on augmentations of the Generative Adversarial Imitation Learning (GAIL) method, are designed to achieve expert-level performance even with confounded data.

**Strengths:**

1. **Interesting Problem:** The paper studies the partial identification problem in imitation learning in sequential decision making. This problem is interesting and has not been investigated before.
2. **Systematic Theoretical Investigations**: The paper conducts systematic theoretical investigations on the partial identification problem in imitation learning, including three cases: non-identifiable transition and reward, identifiable transition and non-identifiable reward, non-identifiable transition and identifiable reward. The theoretical results demonstrate the infeasibility in the fully identifiable setting and the feasibility of the proposed approach in partially identifiable settings.

**Weaknesses:**

1. **Assumptions and Generalization**: The approach relies on certain assumptions about the partial identifiability of the MDP. In practice, these assumptions might not always hold. Even if these assumptions hold, one may not know the type of partial identification in advance, potentially limiting the generalizability of the methods.
2. **Complexity and Practicality**: The proposed method CAIL-$\mathcal{T}$ requires solving a complex constrained optimization problem in Eq. (18) and Eq. (19). It is difficult to solve this optimization problem in practice when neural networks are employed.
3. **Missing Experimental Details:** The paper does not provide details about the implementation of the proposed algorithm and the environments. Besides, the source code is also missing. As such, it is difficult to reproduce their experiments.

**Questions:**

Please see the detailed review in the Weakness part.

**Limitations:**

The authors have thoroughly discussed the limitations of this paper.

---

> ### Author Rebuttal · Authors · 2024-08-07
>
> We sincerely thank your reviewers and recognize that certain elements of our work  might have been misunderstood, which could have influenced the evaluations. Below, we aim to clarify these aspects and remain eager to engage in further dialogue to resolve any lingering doubts. Generally, we have validated the proposed algorithms across a broad range of practical scenarios, including SCMs with various reward functions and transition dynamics, driving, and healthcare, where the proposed algorithms have demonstrated significant improvements in decision-making outcome performance.
>
> ---
> > _“Assumptions and Generalization: The approach relies on certain assumptions about the partial identifiability of the MDP. In practice, these assumptions might not always hold. Even if these assumptions hold, one may not know the type of partial identification in advance, potentially limiting the generalizability of the methods.”_
>
> Standard imitation learning in MDPs assumes both the Markov property (Assumption 1) and Causal consistency (Assumption 2) in the demonstration data. In this paper, we generalize the second assumption as we allow the presence of unobserved confounding or violation of overlap. This means that our proposed methods are applicable to all standard MDP instances, and also generalize well to confounded MDPs where standard imitation learning methods do not necessarily obtain a valid solution. Given the wide adoption of MDPs and imitation learning, and the prevalence of unobserved confounding bias, we strongly believe our proposed method could be applied in many real-world scenarios, e.g., autonomous driving and healthcare.
>
> Table 1 provides a recipe for applying our methods in practice, given the violations of assumptions. First, when Assumption 2 holds and there is no unobserved confounder, the standard imitation learning algorithm obtains an effective policy. When there exist unobserved confounders affecting both the reward function and transition distribution (i.e., both equations in Assumption 2 do not hold), the expert performance is not imitable and the learner should explore additional domain knowledge. When unobserved confounders only affect either the reward function or transition distribution, the imitator could apply our proposed method and obtain a robust policy with performance guarantee.
>
> ---
> > _“Complexity and Practicality: The proposed method CAIL-T requires solving a complex constrained optimization problem in Eq. (18) and Eq. (19). It is difficult to solve this optimization problem in practice when neural networks are employed.”_
>
> We have developed specific methodologies to address these challenges. As outlined in Alg. 3 (see Appendix) and further elaborated in Thm. 3, our approach employs a structured algorithm designed to effectively navigate the optimization landscape. Specifically, as stated in L566-568, “The intuition for Alg. 3 is: in order to find the worst case, we need to put as less transition probability mass as possible to the state with maximal values, and allocate higher transition probabilities to states with smaller values.” Moreover, with some approximation techniques (Xia et al. (2023)), it is possible to solve it with NNs.
>
>
> To substantiate the practical applicability of our methods, we have conducted experiments focusing on Driving scenarios (L300-312). These experiments, detailed in Appendix C, successfully implement the proposed framework using neural networks and demonstrate its effectiveness and efficiency in real-world settings. These results underscore our method's viability and its capability to handle the complexities associated with neural network-based implementations.
>
>
> ---
> > _“Missing Experimental Details: The paper does not provide details about the implementation of the proposed algorithm and the environments. Besides, the source code is also missing. As such, it is difficult to reproduce their experiments.”_
>
> We would like to clarify that our paper has indeed included a comprehensive description of the theoretical framework, algorithms, and step-by-step procedures required to understand and implement the proposed method. Theoretical contributions are summarized in Sec. 2 and 3, and further elaborated in Appendix A and B. Experimental and implementation details are provided in Section 4 and further elaborated in the supplementary materials (Appendix C). These environments are common in the causal imitation learning literature, ensuring that our experiments could be easily replicated. As mentioned in the checklist, we will open-source our codebase upon acceptance of our paper.

---

> > ### Comment · Reviewer_S4Sm · 2024-08-13
> >
> > Thanks a lot for the detailed responses.
> >
> > Regarding the first question, it seems that the response from the authors does not address my concern. My concern here is that, in practice, we typically do not know in advance which type of assumption the underlying task satisfies. For instance, whether the underlying reward function is identifiable or not. In that case, it remains unclear how to choose their methods to solve the task.

---

> > > ### Author Response · Authors · 2024-08-13
> > >
> > > We thank the reviewer for the response. Since the discussion phase is coming to an end, we would like to summarize below the logic behind our model assumptions and how they compare to standard imitation learning methods.
> > >
> > > 1. Standard imitation learning: Markov Property (Definition 1) + Unconofundedness (Definition 2);
> > > 2. Our approach: Markov Property (Definition 1).
> > >
> > > As described above, our model only relaxes existing assumptions in the standard setting and, therefore, is more general, subsuming the standard imitation learning. When the imitator has strong knowledge about the MDP environment and is confident about unconfoundedness, it should follow the standard imitation algorithms and obtain a solution. When the imitator is unsure about the unconfoundedness assumption, it could still apply our method and obtain a robust policy with a performance guarantee. Given the wide applications of existing imitation learning methods and the relaxation of the critical assumption of unconfoundedness, our results could be generalized to many practical domains.

---

### Official Review · Reviewer_1FKA · 2024-07-12

**Soundness:** 3
**Presentation:** 2
**Contribution:** 3
**Rating:** 4
**Confidence:** 4

**Summary:**

The paper addresses the challenges in imitation learning when expert demonstrations are contaminated with unobserved confounding bias. It proposes robust imitation learning methods within the framework of MDPs using partial identification techniques. The authors demonstrate theoretically that when unobserved confounders exist, learning a robust policy to achieve expert performance is generally infeasible. They introduce two novel causal imitation algorithms to handle settings where either the transition distribution or the reward function is non-identifiable from the available data. The proposed methods are validated through experiments, showing their effectiveness in achieving expert-like performance in the presence of unobserved confounders.

**Strengths:**

1. The paper tackles the issue of unobserved confounders in imitation learning, which is a significant challenge in practical applications.
2. The paper provides theoretical foundations, demonstrating the infeasibility of achieving expert performance with unobserved confounders and offering rigorous proofs for the proposed algorithms.
3. The introduction of two novel causal imitation algorithms (CAIL-$\mathcal{R}$ and CAIL-$\mathcal{T}$) enhances the robustness of policy learning in settings where either the transition distribution or the reward function is non-identifiable.

**Weaknesses:**

The experiment evaluation seems insufficient considering that there are already some causal imitation learning baselines such as [1], and some more experiment environments such as OpenAI gym for imitation learning.

[1] P. de Haan, D. Jayaraman, and S. Levine. Causal confusion in imitation learning. In Advances in Neural Information Processing Systems, pages 11693–11704, 2019.

**Questions:**

On what tasks are these approaches experimented on, what do these tasks look like, and how does the demonstration data look like? Will the model performance change with a varying number of expert demonstrations?

**Limitations:**

The limitations have been discussed in the paper.

---

> ### Author Rebuttal · Authors · 2024-08-07
>
> We appreciate your reviews and acknowledge that some aspects of our work might have been misunderstood. Below, we provide clarifications and are open to further discussions should there be any residual concerns.
>
> Our experimental design incorporated similar baselines from Zhang et al. (2020), Kumor et al. (2021), and Ruan et al. (2023), chosen for their direct relevance to our framework. The rationale behind this selection was to showcase the potential of partial identification techniques in improving algorithms such as GAIL. These enhancements are particularly important for achieving expert-level performance despite differences in the observation spaces of the imitator and expert—an aspect often overlooked in OpenAI Gym environments. Most current setups in these environments presuppose identical observation spaces and ignore the presence of unobserved variables, both of which are addressed in our study.
>
>
> ---
> > _“The experiment evaluation seems insufficient considering that there are already some causal imitation learning baselines such as [1], and some more experiment environments such as OpenAI gym for imitation learning.”_
>
> (De Haan et al., 2019) explores the underlying causal relationships in the environment, and exploits the sparsity in these relationships to improve the imitator’s performance. However, their problem setting assumes there is no unobserved confounding in the environment, which is the very motivation and the starting point of this paper. That is, our paper studies the challenges of unobserved confounding for imitation learning in MDPs, which is orthogonal to the setting in (De Haan et al., 2019). Consequently, we did not include (De Haan et al., 2019) in the experiments since their methods and ours are not comparable.
>
>
> We did investigate some existing benchmarks including the OpenAI gym. Unfortunately, as far as we are aware of, no major RL benchmark simulates the presence of unobserved confounding without violating the Markov property. The corresponding causal diagrams of experts are shown in Fig.2 and Fig. 3. Due to these limitations, we have to build novel simulation environments to evaluate our proposed algorithms. Indeed, we intend to release our experiments and hope it could contribute to the existing RL benchmarks by highlighting the challenges of unobserved confounding. We believe that building our own benchmark should be taken as a strength, not a weakness.
>
> ---
> > _”On what tasks are these approaches experimented on, what do these tasks look like, and how does the demonstration data look like? Will the model performance change with a varying number of expert demonstrations?”_
>
> Our learning task is similar to the standard imitation learning setting, where the imitator has access to offline demonstration data generated by an expert. The demonstration data is represented as a collection of sequences of trajectories in an MDP. For every sequence, it contains a finite number of state-action pairs $(s_j, x_j)$; the reward signal $y_j$ is not observed. The main difference is that we consider settings where the demonstration data **could be contaminated with confounding bias**; unobserved confounders exist affecting the action $X_j$,  subsequent state $S_{j+1}$, and reward $Y_j$ (had it been observed).
>
> Specifically, as shown in previous research (Ruan et al. 2022), there exist lots of unobserved variables in real-world human driving datasets, e.g., road conditions. As stated in the driving experiment (L300-312), expert demonstrations include vehicle trajectories, “The state $S_t$ contains some critical driving information, e.g., the velocities of the ego vehicle and the leading vehicle and the spatial distance between them. The action $X_{t}$ represents acceleration or deceleration decisions the ego vehicle makes. The unobserved variable $U_{t}$ represents some information accessible to the expert but inaccessible to the imitator, e.g. slippery road conditions [24].” The details of our medical treatment experiments can be found in L313-325.
>
> Due to challenges caused by the confounding bias existing in expert demonstrations, our paper assumes that the imitator has access to sufficiently much demonstration data. We acknowledge that quantifying the uncertainty due to finite samples in imitation learning is an exciting problem, but it is beyond the scope of this paper.

---

> ### Comment · Reviewer_1FKA · 2024-08-13
>
> Dear AC,
>
> Thank you for initiating this discussion. I acknowledge the differences in problem settings between this work and prior efforts (e.g., De Haan et al., 2019). However, I maintain that including additional baselines is essential, given that the underlying structural causal model is unknown a priori to the agent. Incorporating more baselines and discussing their relevance will provide a clearer understanding of the performance of the proposed approaches.

---

> ### Comment · Reviewer_1FKA · 2024-08-13
>
> Thank you to the authors for the detailed responses. I do recognize the differences in problem settings between this work and previous studies. However, I still believe it is necessary to include additional baselines, especially since the underlying SCM is not known in advance to the agent. Adding more baselines and discussing their relevance will enhance our understanding of how effectively the proposed approaches perform.

---

### Author Rebuttal · Authors · 2024-08-07

# Overall Response

We appreciate the reviewer’s feedback. We believe that a few misunderstandings of our work led to some of the evaluations being overly harsh and would sincerely ask the reviewers to reconsider our paper given the clarifications provided in the response. We will first address some main comments here and then reply to every reviewer separately.

**Unobserved Variables and Non-identifiability** Unobserved confounders generally exist in demonstrations when the sensory capabilities of the imitator and the expert differ or privacy concerns are serious, e.g., HighD (Krajewski et al., 2018) or MIMIC-III (Johnson et al. 2016).  In general, the structural assumptions (e.g., causal diagrams) required to perform causal inferences are inevitable, as shown in (Bareinboim et al., 2022, Theorem 1). Recognizing non-identifiability as a core challenge in causal inference, our approach improves previous partial identification techniques and applies them to sequential decision making settings.

**Significance and Novelty** The central focus of our paper is on exploring the effects of unobserved confounders in imitation learning (IL), an area that, while touched upon, has not been thoroughly explored in existing frameworks. Unlike previous studies such as de Haan et al. (2019), which assume the absence of unobserved confounders, our paper delves into scenarios where either the transition dynamics, the reward function, or both, are confounded (summarized in Table 1). The proposed framework is a substantial deviation from traditional causal imitation learning approaches which often ignore such complexities.

Furthermore, standard imitation learning in MDPs assumes both the Markov property (Assumption 1) and Causal consistency (Assumption 2) in the demonstration data. In this paper, we generalize the second assumption as we allow the presence of unobserved confounding or violation of overlap. This means that our proposed methods are applicable to all standard MDP instances, and also generalize well to confounded MDPs where standard imitation learning methods do not necessarily obtain a valid solution. Given the wide adoption of MDPs and imitation learning, and the prevalence of unobserved confounding bias, we strongly believe our proposed method could be applied in many real-world scenarios, e.g., autonomous driving and healthcare.

To the best of our knowledge, we are the first to theoretically prove that “when unobserved confounders generally exist, it is _infeasible_ to learn a robust policy that is guaranteed to achieve expert performance from the demonstration data.” Additionally, our comprehensive experiments, which span various reward functions and transition dynamics, support these findings. Such findings are nontrivial, because it helps to explain why in practice lots of imitation learning cannot work very well.

Building on the theoretical foundation, to “recover” from the unobserved confounding bias, we have proposed novel imitation learning algorithms using partial identification techniques (as detailed in Alg. 1 and Alg. 2), allowing the imitator to obtain effective policies that can achieve expert performance for different problem settings. The intuition is to optimize the policy within the worst-case environment compatible with the demonstration data and model assumption. Such strategy helps to enhance the robustness and applicability of imitation learning in complex real-world environments.

**Experiment Baselines** In our experimental setup, we've chosen similar baselines from Zhang et al. (2020), Kumor et al. (2021), and Ruan et al. (2023) due to their direct relevance to our experimental framework. This selection is intended to demonstrate how partial identification techniques can enhance existing algorithms like GAIL, helping them achieve expert performance even when the observation spaces between the imitator and expert are different. This is crucial as most existing OpenAI Gym environments do not account for discrepancies between the observation spaces of the expert and the imitator, and assume unobserved variables do not generally exist, which we address in our experiments.

Overall, our paper introduces significant theoretical and practical contributions to the field of imitation learning, opening up new pathways for research and application in environments affected by unobserved confounders.

---

### Decision · Program_Chairs · 2024-09-25

**Decision:**

Accept (poster)

**Comment:**

The paper studies imitation learning for MDPs. The focus is on using causal methods to battle confounding.

The contributions are:
- an impossibility result about unobserved confounders (Theorem 1)
- a result about how to construct a policy when the transition dynamics are identifiable but the reward is not (Theorem 2)
- a result about how to construct a policy when the reward is identifiable but the transition is not (Theorem 3)

While reviewers have voiced concerns, the meta-reviewer disagrees with their assessment. Specifically:
- the lack of extensive experiments shouldn't be a main reason to reject a theory paper.
- saying it's hard to validate assumptions in practice is a vacuous statement, which applies to (almost) all theory papers, especially to causality papers
- saying the paper isn't practical because it involves constrained optimisation is unfair - constrained optimisation is required by many NeurIPS papers.
- criticisms of reproducibility are exaggerated given the paper is theoretical and authors promised to release code upon acceptance
- some said the paper needs more baselines, but didn't provide what baselines to use (and an earlier suggestion turned out irrelevant)
- similarity in algorithms doesn't mean the theory is trivial. The algorithms are different from GAIL in ways which are crucial for battling confounding.

This paper is a solid contribution and pushes on what is possible in causal imitation learning. For this reason, I recommend acceptance.